

# Describing hadronization via histories and observables
# for Monte-Carlo event reweighting

Christian Bierlich[1⋆], Phil Ilten[2†], Tony Menzo[2‡], Stephen Mrenna[2,3∘], Manuel Szewc[2,4§],
Michael K. Wilkinson[2¶], Ahmed Youssef[2‖] and Jure Zupan[2⊥]

**1** Department of Physics, Lund University, Box 118, SE-221 00 Lund, Sweden
**2** Department of Physics, University of Cincinnati, Cincinnati, Ohio 45221, USA
**3** Scientific Computing Division, Fermilab, Batavia, Illinois, USA
**4** International Center for Advanced Studies (ICAS), ICIFI and ECyT-UNSAM,
25 de Mayo y Francia, (1650) San Martín, Buenos Aires, Argentina

⋆ christian.bierlich@hep.lu.se , † philten@cern.ch , ‡ menzoad@mail.uc.edu ,
∘ mrenna@fnal.gov , § szewcml@ucmail.uc.edu , ¶ michael.wilkinson@uc.edu ,
‖ youssead@ucmail.uc.edu , ⊥ zupanje@ucmail.uc.edu

## Abstract

We introduce a novel method for extracting a fragmentation model directly from experimental data without requiring an explicit parametric form, called Histories and Observables for Monte-Carlo Event Reweighting (HOMER), consisting of three steps: the training of a classifier between simulation and data, the inference of single fragmentation weights, and the calculation of the weight for the full hadronization chain. We illustrate the use of HOMER on a simplified hadronization problem, a $q\bar{q}$ string fragmenting into pions, and extract a modified Lund string fragmentation function $f(z)$. We then demonstrate the use of HOMER on three types of experimental data: (i) binned distributions of high-level observables, (ii) unbinned event-by-event distributions of these observables, and (iii) full particle cloud information. After demonstrating that $f(z)$ can be extracted from data (the inverse of hadronization), we also show that, at least in this limited setup, the fidelity of the extracted $f(z)$ suffers only limited loss when moving from (i) to (ii) to (iii). Public code is available at https://gitlab.com/uchep/mlhad.

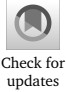

# 1  Introduction

Machine Learning (ML) methods provide a new set of tools that may be able to improve current descriptions of the non-perturbative process of hadronization – the binding of quarks and gluons into observable hadrons. Indeed, the past several years have seen a gradual development of ML approaches to hadronization, notably from the MLHAD [1,2] and HADML [3–5] collaborations. The long-term goal of such efforts is not only to supplement the phenomenological hadronization methods used in the state-of-the-art Monte Carlo simulations such as PYTHIA [6], but to also use these models to better understand the underlying physics of hadronization. This can be particularly useful in cases where hadronization uncertainty plays a critical role in experimental measurements, *e.g.*, the mass of the top quark, or when an accurate subtraction of the underlying event is necessary as is common in heavy ion jet measurements.

In this manuscript, we demonstrate how the symmetric Lund string fragmentation function $f(z)$ can be learned directly from data using ML techniques in the simplified case of a $q\bar{q}$ string fragmenting to pions. We do this using a new strategy, the Histories and Observables for Monte-Carlo Event Reweighting (HOMER) method, which uses phenomenologically-motivated hadronization models, *e.g.*, the Lund string model from PYTHIA, as a starting point. In the HOMER method, one first learns the event-level likelihood ratios between the distributions from data and the chosen hadronization model. These event-level likelihood ratios are then used to build a modified hadronization model by assigning likelihood ratios for each individual hadronization emission. The output of the HOMER method is a data-driven reweighting of the baseline PYTHIA event generator, such that the resulting distributions match the observed training data. In our previous work, ref. [2], we introduced a similar methodology for extracting microscopic dynamics from macroscopic observables relying on the explicit construction of a likelihood using normalizing flows, which also utilized the Lund string model as a starting point. In contrast, the HOMER method does not construct an explicit likelihood function but instead learns the likelihood ratio. This allows for a more straightforward incorporation of the model into the simulation pipeline. HOMER also implements a different training procedure based on a convex loss function, improving convergence during training in comparison to using an adversarial non-convex loss function.

This manuscript is structured as follows. We briefly review the Lund string fragmentation model in section 2.1, present the details of the HOMER method in section 2.2, and compare the HOMER method with Generative-Adversarial-Network (GAN)-based approaches in section 2.3. Numerical results are presented in section 3 for three different examples of available experimental information; in section 3.1.1, binned distributions of high-level observables such as event shapes and particle multiplicities are used, in section 3.1.2 the unbinned measurements of high-level observables on an event-by-event basis are used, and in section 3.2 we explore the case where full particle-cloud information is available. Section 4 contains our conclusions and future outlook. Appendix A contains details of the public code, while for convenience, appendix B defines the shape observables used in section 3.1, and appendix C contains additional numerical results and figures, supplementing the results shown in sections 3.1.2 and 3.2.

## 2 The HOMER method

The HOMER method is a framework to learn a hadronization model from data without requiring an explicit parametric functional form; here, the model is a modified Lund string fragmentation function $f(z)$, described below in section 2.1. We demonstrate the functionality of the HOMER method using synthetic data generated with PYTHIA, which allows us to examine how well the extracted string fragmentation function $f_{\mathrm{HOMER}}(z)$ approximates the actual function used by the synthetic data $f_{\mathrm{data}}(z)$. This actual $f_{\mathrm{data}}(z)$ is not available in data, even synthetic data, where the ordering of the hadron emissions cannot be measured and leads to an ambiguity in the possible fragmentation chains that could produce an observable event.

The starting point of HOMER is a simulated hadronization model, *e.g.*, PYTHIA with a reasonable set of parameters, which is assumed to already give a decent approximation of the data. HOMER then uses data in a two step procedure to transform this **baseline simulation model** to match the data. In our case, the simulator is PYTHIA, but with the initial string fragmentation function, $f_{\mathrm{sim}}(z)$, using different parameters than those used to generate the synthetic data, $f_{\mathrm{data}}(z)$. The simulator produces **events**, which can then be compared to data. In our terminology, an event $e_i$ is a list of observables, $\vec{x}_i$, which describe a *single collision*, *e.g.*, a single instance of $e^+e^- \to u\bar{u}$ annihilation. A collection of events $\{\vec{x}_1, \ldots, \vec{x}_N\}$ is called a **run**. For observables $\vec{x}_i$ we consider two possibilities:

   *i.* *high-level observables* (section 3.1) constructed from particle level information, such as thrust, multiplicity, *etc.* and

   *ii.* *point cloud* (section 3.2), in which case $\vec{x}_i$ contains the four momenta of all hadrons.

In the following sections, we review the Lund string fragmentation model and give further details about the HOMER method.

### 2.1 Lund string fragmentation model for the $q\bar{q}$ case

The hadronization model in PYTHIA is the Lund string fragmentation model [7, 8]. PYTHIA is a multi-purpose Monte Carlo event generator which can simulate particle collisions for a wide range of initial states, including proton-proton and heavy-ion collisions. A PYTHIA event begins with the production of a hard partonic process, followed by the application of a parton shower, underlying event production, and finally hadronization of the partons and particle decays. Here we limit our discussion to the simplest hadronizing partonic system: a pair of massless first generation quarks with flavor $i$, $q_i\bar{q}_i$, with no gluons attached and fragmenting only into pions. The addition of gluons will be explored in future work [9]. The momenta of the quarks are taken to be oriented along the $z$ axis, and the quark flavors are $u = q_1$ and

$d = q_2$. In the Lund model, an approximately uniform flux tube of color field, a massless relativistic string with tension $\kappa \approx 1\,\text{GeV/fm}$, forms when the quark and antiquark become spatially separated. The two endpoints of the string are thus the quark and antiquark. This state will decay into multiple hadrons. In a simple model with one spatial dimension and only one quark flavor, the probability for a state with $n$ mesons with momenta $p_i = \{1, 2, ..., n\}$ is given by [7]:

$$\mathcal{P} \propto \left\{ \left[ \prod_1^n N d^2 p_i \delta\left(p_i^2 - m^2\right) \right] \delta^{(2)}\left(\sum p_i - P_{tot}\right) \right\} \exp(-bA), \tag{1}$$

where the term $bA$ corresponds to the imaginary part of the action of a massless string. The area $A$ is the space-time area of the string scaled by $\kappa^2$. Generating the transition from a string state to a hadronic state which fulfills eq. (1), is now a question of selecting a specific implementation. In the Lund model, the increasing separation between the quark and the antiquark, makes it energetically favorable to create $q_j \bar{q}_j$ pairs out of the vacuum. The string therefore breaks into spacelike separated fragments, *i.e.*, hadrons. Due to the spacelike separation, there is no preferred time-ordering, and the hadronization process could therefore be described either by an inside-out cascade, starting from the center and fragmenting outwards, or an outside-in cascade, starting from the string ends. The Lund model makes the latter selection. Starting from one string end, a string break now produces a hadron containing the string end quark (or anti-quark), and the anti-quark (or quark) from the string break. In this way, the $j$'th string break will produce the hadron containing $q_{j-1}\bar{q}_j$ or vice versa. We call this sequence of multiple hadron emissions from a single string fragment a **fragmentation chain**.

Each string break is treated probabilistically. After flavor selection, the transverse momentum of the $q_j \bar{q}_j$ pair, $\Delta \vec{p}_T = (\Delta p_x, \Delta p_y)$, is sampled from a phenomenological normal distribution that has an adjustable width of $\sigma_\text{T}/\sqrt{2}$. The emitted hadron is given longitudinal lightcone momentum, by taking away a fraction $z$ of the remaining lightcone momentum of the string, defined as

$$z \equiv (E \pm p_z)_\text{had} / (E \pm p_z)_\text{string}, \tag{2}$$

where $E$ and $p_z$ are the energy and longitudinal momentum of the hadron or string, as labeled, and the $+$ $(-)$ sign corresponds to the string break occurring at the $+\hat{z}$ $(-\hat{z})$ end of the string. This updates the remaining light-cone momentum of the string for the next iteration. The value of $z$ is sampled from the symmetric Lund fragmentation function

$$f(z) \propto \frac{(1-z)^a}{z} \exp\left(-\frac{bm_\text{T}^2}{z}\right), \tag{3}$$

where $m_\text{T}^2 \equiv m_{ij}^2 + p_\text{T}^2$ is the square of the transverse mass, $m_{ij}$ is the mass of the emitted hadron, and $a$ and $b$ are fixed parameters determined from fits to data. Note that in eq. (3) we do not include a normalization factor to make this a true probability distribution.[1] Each iteration of causally disconnected string fragmentations consists of:

1. randomly selecting one string end;

2. assigning probabilistically a quark flavor to be pair produced during string breaking;

3. selecting the corresponding hadron defined by its mass and spin, in other words, resolving the system as a hadron;

4. generating the transverse momentum of this pair;

---

[1]In PYTHIA, samples from $f(z)$ are obtained via an accept-reject algorithm where only the location of its maximum, which can be calculated analytically, is needed.

5. generating the light-cone momentum fraction of the new hadron;

6. and finally computing the energy and longitudinal momentum of the new hadron from eq. (2) and

$$(E - p_z)_{\text{had}}(E + p_z)_{\text{had}} = m_{\text{T}}^2 .$$

The transverse momentum of the emitted hadron, $\vec{p}_{\text{T}}$, is constructed as the combined $\vec{p}_{\text{T}}$ of the two quarks entering it. If the hadron is the result of two neighboring string breaks $i$ and $j$, then $\vec{p}_{\text{T}}$ is the (vector) sum of $\vec{p}_{\text{T},i}$ and $\vec{p}_{\text{T},j}$. The end-point hadrons contain the endpoint quarks, which have no $\vec{p}_{\text{T}}$ (the simulation of hadronization takes place in the string rest frame).

Since hadron masses are discrete and the fragmentation function in eq. (3) carries no information about the global state of the string, care must be taken near the end of the iterative process to produce a physical state. When the remaining string system has an invariant mass below a chosen low-energy threshold, it is then converted into a final pair of hadrons. In PYTHIA, this conversion is performed by the `finalTwo` algorithm. The `finalTwo` method effectively works as a filter by checking whether the final hadrons can be produced on-shell while ensuring that the overall fragmentations follow the left-right symmetric Lund fragmentation function. If this is not the possible, the **entire** generated fragmentation chain is rejected, and the simulation of hadronization starts anew, taking again the original string as the starting point. This process can be repeated several times until the `finalTwo` step is successful.

The effect of `finalTwo` is that the **observable event** does not contain enough information to reconstruct the full **internal simulation history**[2] since rejected chains are not stored in the PYTHIA event record. In experimental data, the observable event contains all the particles measured by the detector for a single collision.[3] Our simulated data from PYTHIA is the ideal case where all of the final-state particles produced by the event generator are detectable. The simulated data suffers from the same problem that it does not include enough information to describe the internal simulation history. In general, it is uncommon to retain information about states that were rejected, because the amount of data from an inefficient filter would be prodigious.[4] To build the necessary internal simulation history, we supply a custom `UserHooks` object[5] to PYTHIA. This distinction between the observable event and internal simulation history will be important to keep in the HOMER setup, in order to faithfully re-interpret PYTHIA simulated data, so that the **simulated observable event** becomes statistically indistinguishable from the **measured observable event**.

String breaks are the production points of hadrons in the hadronization process. They are described by seven dimensional vectors containing: the light-cone momentum fraction of the hadron, $z$; the two-dimensional momentum kick $\Delta\vec{p}_{\text{T}} = (\Delta p_x, \Delta p_y)$ of the emitted hadron; the hadron mass $m_{ij}$; a boolean that encodes whether the string break occurred at the positive or the negative end of the string, stored within PYTHIA as `fromPos`; and the transverse momentum of the string before the breaking, $\vec{p}_{\text{T}}^{\text{string}} = (p_x^{\text{string}}, p_y^{\text{string}})$[6]

$$\vec{s}_{hcb} = \{z, \Delta\vec{p}_{\text{T}}, m, \texttt{fromPos}, \vec{p}_{\text{T}}^{\text{string}}\}_{h,c,b} . \tag{4}$$

---

[2]It is possible to reconstruct $\vec{p}_{\text{T}}$, flavour and the full string area *c.f.* eq. (1), but not the ordering.

[3]This neglects detector effects such as efficiency and resolution which degrade the observable event.

[4]In principle, all of the details of the generation can be reconstructed from the algorithm and random number seed.

[5]A `UserHooks` object allows the person running the program, a `User`, to `Hook` into the event generation process and access internal variables normally not visible.

[6]In general, the energy of the string in its center-of-mass frame, $E_{\text{string}}$, would also be included in this list. We are able to use the shorter list in eq. (4), since the fragmentations in the Lund string model do not depend explicitly on $E_{\text{string}}$. Additional information must also be included in the case where gluons are attached to the string, and the initial state changes between events.

The indices $h$, $c$, and $b$ indicate the history, fragmentation chain, and string break the vector belongs to, as defined below. Note that $\vec{s}_{hcb}$ contains redundant information, since the transverse momentum of the string fragment after the hadron emission, $(\vec{p}_{\mathrm{T}}^{\,\mathrm{string}})_{hcb}$, can be reconstructed from the $(\Delta\vec{p}_{\mathrm{T}})_{hcb}$ of all previous hadron emissions. However, we find it convenient to keep $\vec{p}_{\mathrm{T}}^{\,\mathrm{string}}$ explicitly as a datum in $\vec{s}_{hcb}$.

A sequence of string breaks forms a fragmentation chain,

$$\vec{S}_{hc} = \{\vec{s}_{hc1}, \ldots, \vec{s}_{hcN_{h,c}}\}, \tag{5}$$

while a vector of rejected fragmentation chains and the accepted fragmentation chain form a fragmentation history,[7]

$$\vec{\mathbf{S}}_h = \{\vec{S}_{h1}, \ldots, \vec{S}_{hN_h}\}. \tag{6}$$

Here, the indices are defined as: $h = 1, \ldots, N_{\mathrm{data}}$ is the fragmentation history index, with $N_{\mathrm{data}}$ the total number of events in a run; $c = 1, \ldots, N_h$ is the fragmentation chain index for a particular fragmentation history $h$, which has $N_h - 1$ rejected fragmentation chains and one accepted fragmentation chain; and $b = 1, \ldots, N_{h,c}$ is the string break index, that runs over the fragmentation chain $c$ with a total of $N_{h,c}$ string breaks. The form of eq. (6) assumes the simplified scenario of this work, where only one string is hadronized per event. When there are multiple strings per event then the fragmentation history is simply expanded to be the vector of accepted and rejected fragmentation chains for all strings, including the energy of each string.

In summary, a measurable **event** $e_h$, where the index $h = 1, \ldots, N_{\mathrm{data}}$ runs over all events in a run, is fully described by specifying the accepted fragmentation chain

$$e_h \equiv e(\vec{\mathbf{S}}_h) \equiv e(\vec{S}_{hN_h}). \tag{7}$$

Explicitly, $e_h$ is an unordered list of $N_{\mathrm{had}} = N_{h,N_h} + 2$ laboratory frame four momenta, $(E_i, \vec{p}_i)$, and masses, $m_i$, of the produced hadrons,[8]

$$e_h = \{\{m_{h1}, E_{h1}, \vec{p}_{h1}\}, \ldots, \{m_{hN_{\mathrm{had}}}, E_{hN_{\mathrm{had}}}, \vec{p}_{hN_{\mathrm{had}}}\}\}. \tag{8}$$

This unordered list is constructed from the accepted fragmentation chain quantities, $\vec{S}_{hN_h}$, by boosting the momenta of the produced hadrons to the laboratory frame. If two simulation histories differ only by their rejected chains, they result in the same event and are equal to the event given by the accepted fragmentation chain. An example schematic of all the components for a fragmentation history, and run, are shown in fig. 1. In this work here, it is important to note that our synthetic PYTHIA data, unlike real data, also contain the rejected fragmentation chains, so that the fragmentation history contains a vector $\{\vec{S}_{h1}, \ldots, \vec{S}_{hN_h}\}$ for each event $e_h$. This is not relevant for the HOMER method, but allows us to perform a closure test of the method.

## 2.2 Details about the HOMER method

In the Lund string fragmentation model, the probability of a given string break $\vec{s}_{hcb}$ depends on the $\vec{p}_{\mathrm{T}}^{\,\mathrm{string}}$ of the string fragment that is emitting the hadron. We write this conditional

---

[7]For the accepted fragmentation chain, `finalTwo` proceeds to generate two additional hadrons from the remaining string. Thus, the total number of hadrons will be the number of string breaks contained in the accepted chain plus the additional two hadrons, which are not counted as string breaks in this work. This is because their kinematic distribution does not depend on the $a$ and $b$ Lund parameters as long as the low-energy threshold that determines whether `finalTwo` is applied remains fixed, and thus do not need to be reweighted for the example we consider here.

[8]Additional information could be optionally included for each hadron such as flavor composition or charge.

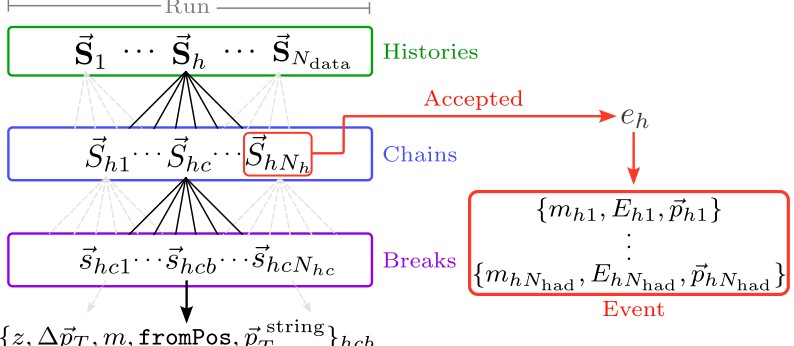

Figure 1: Schematic detailing the different components of a simulated run. String breaks are grouped into fragmentation chains, while collections of rejected and accepted fragmentation chains form fragmentation histories. Observable events are obtained from the last, accepted fragmentation chain. A collection of multiple events is a run.

probability as

$$p(\vec{s}_{hcb}) \equiv p(\{z, \Delta\vec{p}_{\mathrm{T}}, m, \texttt{fromPos}\}_{h,c,b} | \{\vec{p}_{\mathrm{T}}^{\ \mathrm{string}}\}_{h,c,b}). \tag{9}$$

HOMER aims to learn a fragmentation weight $w_{\mathrm{s}}^{\mathrm{data}}(\vec{s}_{hcb})$ for individual hadron emissions. This is achieved by taking an initial guess for the fragmentation probability function $p_{\mathrm{sim}}(\vec{s}_{hcb})$ from the **baseline simulation model**, *e.g.*, PYTHIA, and reweighting it to a data-driven fragmentation function

$$p_{\mathrm{sim}}(\vec{s}_{hcb}) \rightarrow w_{\mathrm{s}}^{\mathrm{infer}}(\vec{s}_{hcb}) p_{\mathrm{sim}}(\vec{s}_{hcb}). \tag{10}$$

In this work, we make the simplifying assumption that only pions are produced in fragmentation, leading to a simpler transverse mass spectrum (the exploration of more realistic scenarios, including data-driven flavor selection is left for future work). The baseline simulation model is taken as eq. (3), with parameters chosen to differ sufficiently from those used to produce the synthetic data. $N_{\mathrm{sim}}$ baseline simulation model events are generated. The baseline simulation model events are then compared with the experimental run dataset that contains $N_{\mathrm{data}}$ events. In obtaining $w_{\mathrm{s}}^{\mathrm{infer}}(\vec{s}_{hcb})$, HOMER only uses information that is experimentally accessible.

In the HOMER method, the inference process is divided into three steps. In step 1 the ratio of probabilities for a given event $e_h$ to occur in data compared to the baseline simulation model,

$$w_{\mathrm{exact}}(e_h) = \frac{p_{\mathrm{data}}(e_h)}{p_{\mathrm{sim}}(e_h)}, \tag{11}$$

is estimated with a classifier, $w_{\mathrm{class}}(e_h) \approx w_{\mathrm{exact}}(e_h)$. The classifier weights of step 1 are then used in step 2 to infer single emission weights, $w_{\mathrm{s}}^{\mathrm{infer}}(\vec{s}_{hcb})$. These single emission weights are finally combined in step 3 to give the predicted weight for a fragmentation history, $w_{\mathrm{HOMER}}(\vec{\mathbf{S}}_h)$. In the following, we provide details for each of these three steps.

Since the data used in this work is synthetic data produced with PYTHIA, the exact weights, $w_{\mathrm{exact}}(e_h)$, are known. These weights are controlled by the ratio of probabilities for an event produced either using the "sim" or "data" values of parameters for the Lund fragmentation function of eq. (3); the corresponding fragmentation functions for these two parameter sets are denoted as $f_{\mathrm{sim}}(z)$ and $f_{\mathrm{data}}(z)$, respectively. For a single emission the exact weight is then simply given by

$$w_{\mathrm{s}}^{\mathrm{exact}}(\vec{s}_{hcb}) = \frac{f_{\mathrm{data}}(z)}{f_{\mathrm{sim}}(z)}, \tag{12}$$

and thus is a known analytic function, up to the normalization constants for each $f(z)$ (see ref. [10] for details). The exact weights for an event, $w_{\text{exact}}(e_h)$, are then built from the exact single emission weights $w_s^{\text{exact}}(\vec{s}_{hcb})$.

### 2.2.1 Step 1: Event classifier

To estimate $w_{\text{exact}}(e_h)$, we train a Machine Learning (ML) algorithm to distinguish between data and the events produced from the baseline simulation model. This classifier can only have access to measurable quantities – the kinematic and flavor information for the observable hadrons in each event – and not to the full fragmentation history. In practice, the experimental measurements that can be made realistically now or in the near future never use this full information, but, rather, collections of high-level observables. To study the performance of HOMER with such observables, we consider two limits: **unbinned** and **binned** scenarios. The unbinned scenario requires dedicated experimental measurements in the same vein as ref. [11], while the binned scenario uses only information that is already available from LEP measurements archived on HEPDATA [12]. We also benchmark the performance of our model when using the full information available, a **point cloud** representation of events. Such point cloud datasets will be available from experiments either as full phase-space measurements unfolded using techniques such as OMNIFOLD [13–15], or through open-data initiatives that will still require detector simulations and performance maps to correct reconstruction-level data.

**The unbinned scenario.** In this scenario, the information available for an event consists of values for observables $\mathcal{O}_i$, such as the charged multiplicity, thrust, *etc.*, for every event. A full list of the $N_{\text{obs}} = 13$ observables we use is given in section 3.1. More precisely, in this scenario the information for each event $e_h$ is given by a vector

$$\vec{x}_h = \{\mathcal{O}_1(e_h), \ldots, \mathcal{O}_{N_{\text{obs}}}(e_h)\} . \tag{13}$$

The available information for an experimental run is therefore represented by a collection of all $\vec{x}_h$, giving the input vector $\vec{X}_{\text{data}} = \{\vec{x}_1, \ldots, \vec{x}_{N_{\text{data}}}\}$. Similarly, the $N_{\text{sim}}$ baseline simulation model events are collected in the baseline simulation model input vector, $\vec{X}_{\text{sim}} = \{\vec{x}_1, \ldots, \vec{x}_{N_{\text{sim}}}\}$.

The step 1 classifier is trained using the standard Binary Cross-Entropy (BCE) loss function, which for balanced classes is

$$\mathcal{L}_{\text{unbin}} = -\frac{1}{N_{\text{sim}}} \sum_{h=1}^{N_{\text{sim}}} \ln\left(1 - y(\vec{x}_h)\right) - \frac{1}{N_{\text{data}}} \sum_{h=1}^{N_{\text{data}}} \ln\left(y(\vec{x}_h)\right) , \tag{14}$$

where $y(\vec{x}_h) \in [0, 1]$ is the output of a classifier for the input vector $\vec{x}_h$, and $N_{\text{sim}}$ and $N_{\text{data}}$ are the number of training samples per class, which we assume to be equal in eq. (14). If needed, we weight the classes to ensure that each class has the same weighted number of events. This choice of the loss function guarantees that when the classifier training converges, we can obtain a good estimator for the event weight of eq. (11) from the output of the classifier,

$$w_{\text{class}}(e_h) \equiv \frac{y(\vec{x}_h)}{1 - y(\vec{x}_h)} , \tag{15}$$

so that $w_{\text{exact}}(e_h) \approx w_{\text{class}}(e_h)$.

**The binned scenario.** Here, the classifier input is still the vectors of observables for each event, $\vec{x}_h$ of eq. (13). However, measured data is partitioned into disjoint bins $n_i$ of observable

$\mathcal{O}_i$ values. The classifier is trained using a loss function constructed from the binned measured data and the reweighted baseline simulation model,

$$\mathcal{L}_{\text{bin}} = \sum_{\mathcal{O}_i} \frac{N_{\text{data}}}{n_i} \sum_{k=1}^{n_i} \frac{\left(p_k^{\mathcal{O}_i} - \bar{p}_k^{\mathcal{O}_i}(y)\right)^2}{p_k^{\mathcal{O}_i}} , \tag{16}$$

where the summation is over all the observables up to index $N_{\text{obs}}$. The classifier attempts to minimize the difference between the measured distributions and the reweighted distributions of the baseline simulation model. That is, for each observable $\mathcal{O}_i$, the fractions of events $p_k^{\mathcal{O}_i}$ in a particular bin are given by,

$$\left\{p_1^{\mathcal{O}_i}, \dots, p_{n_i}^{\mathcal{O}_i}\right\} = \frac{1}{N_{\text{data}}} \left\{N_1^{\text{data}}\big|_{\mathcal{O}_i}, \dots, N_{n_i}^{\text{data}}\big|_{\mathcal{O}_i}\right\} , \tag{17}$$

and similarly for the expected fractions of events, $\bar{p}_k^{\mathcal{O}_i}$, estimated from the baseline simulation model. For this, each event $e_h$ is weighted with $w_{\text{class}}(e_h) = y(\vec{x}_h)/\left(1 - y(\vec{x}_h)\right)$, so that $\mathcal{L}_{\text{bin}}$ is minimized for $w_{\text{class}}(e_h) \approx w_{\text{exact}}(e_h)$. For instance, if the observable $\mathcal{O}_i$ is the charged multiplicity, $n_{\text{ch}}$, then $N_1^{\text{data}}|_{n_{\text{ch}}}$ gives the number of events in the run that have $n_{\text{ch}} = 2$, $N_2^{\text{data}}|_{n_{\text{ch}}}$ the number of events in the run that have $n_{\text{ch}} = 4$, *etc.*, and similarly for simulation, but now for weighted distributions.

By construction, the loss function $\mathcal{L}_{\text{bin}}$ in eq. (16) again guarantees that the output of a converged classifier can be used to approximate the event weights of eq. (11), using the output of the classifier from eq. (15). The use of classifiers for reweighting simulated events so that their distributions match the measured event distributions is a common technique (see, *e.g.*, refs. [16–19]). The novelty of the current approach is in establishing a relationship between event weights and the underlying fragmentation function, made possible via the application of eq. (23). This is exploited in step 2 to infer an estimator for $w_s$.

**The Point cloud scenario.** For this scenario we represent an event $e_h$ as a vector $\vec{x}_h$ of four momenta, see eq. (8). $\vec{x}_h$ contains all the information available in the event except particle type and charge. This is the limiting case of the unbinned scenario and therefore proceeds analogously; the available information about the experimental run is represented by a collection of all $\vec{x}_h$, giving the input vector $\vec{X}_{\text{data}} = \{\vec{x}_1, \dots, \vec{x}_{N_{\text{data}}}\}$. The $N_{\text{sim}}$ baseline simulation model events are collected in the baseline simulation model input vector, $\vec{X}_{\text{sim}} = \{\vec{x}_1, \dots, \vec{x}_{N_{\text{sim}}}\}$. As in the unbinned scenario, a classifier is trained using the standard BCE loss function of eq. (14), and we obtain $w_{\text{class}}(e_h) \approx w_{\text{exact}}(e_h)$ through eq. (15).

### 2.2.2 Step 2: Inference of fragmentation weights

The goal of this step is to construct the appropriate single emission weights $w_s$ such that the probability for each string break that produces the emission, $p_{\text{data}}(\vec{s}_{hcb})$ of eq. (10), will reproduce data by reweighting the baseline simulation model string breaks. However, there are two complications. First, in PYTHIA, the baseline fragmentation history also contains the string fragmentation chains that do not pass the `finalTwo` filter (as explained in section 2.1). That is, the `finalTwo` filter divides the fragmentation chains $\{\vec{S}_{hc}\}$ in two: chains that pass the `finalTwo` filter, a set consisting of $\{\vec{S}_{hc}^{\text{acc}}\}$, and those that do not, set $\{\vec{S}_{hc}^{\text{rej}}\}$. Second, the measurable event quantities, *i.e.*, the momenta of the outgoing hadrons of $e_h$, given by eq. (8), form an unordered list, since there is no information about the sequence of causally disconnected string breaks. This means that two fragmentation chains, $\vec{S}_{hN_h}$ and $\vec{S}'_{hN_h}$, which give rise

to exactly the same hadron four momenta except with a different order of emissions, are physically indistinguishable. That is, the two fragmentation chains give rise to the same observable event, $e_h = e(\vec{S}_{hN_h}) = e(\vec{S}'_{hN_h})$.

The probability for an event in the baseline simulation model is thus given by

$$
\begin{aligned}
p_{\text{sim}}(e_h) &= \left( \sum_{N_h=1}^{\infty} \left( p_{\text{sim}}^{\text{rej}} \right)^{N_h-1} \right) \times \left( \sum_{e(\vec{S}_{hN_h})=e_h} p_{\text{sim}}(\vec{S}_{hN_h}) \right) \\
&= \frac{1}{1 - p_{\text{sim}}^{\text{rej}}} \sum_{e(\vec{S}_{hN_h})=e_h} p_{\text{sim}}(\vec{S}_{hN_h}) = \frac{1}{p_{\text{sim}}^{\text{acc}}} \sum_{e(\vec{S}_{hN_h})=e_h} p_{\text{sim}}(\vec{S}_{hN_h}),
\end{aligned}
\tag{18}
$$

where the second summation of the first line is over the set of accepted fragmentation chains that lead to the same observable event $e_h$. The total probability of producing the event $e_h$ also contains the probability of rejecting fragmentation chains. Since the specifics of the rejected chains do not matter, as they are statistically independent of the accepted chain, the accepted chain has no dependence on past rejected chain(s) and the probability that enters $p_{\text{sim}}(e_h)$ is the probability of rejecting *any* chain. This is given by summing over the set of rejected chains,

$$
p_{\text{sim}}^{\text{rej}} = \sum_{\vec{S}_{jk} \in \{\vec{S}_{hc}^{\text{rej}}\}} p_{\text{sim}}(\vec{S}_{jk}),
\tag{19}
$$

where the summation over $N_h$ in eq. (18) counts the number of fragmentation chains that are rejected in the simulation before $\vec{S}_{hN_h}$ is accepted.[9] Similarly, the total probability for the accepted fragmentation chains is given by

$$
p_{\text{sim}}^{\text{acc}} = 1 - p_{\text{sim}}^{\text{rej}} = \sum_{\vec{S}_{jk} \in \{\vec{S}_{hc}^{\text{acc}}\}} p_{\text{sim}}(\vec{S}_{jk}),
\tag{20}
$$

Note that the probabilities for individual fragmentation chains are products of string breaks,

$$
p_{\text{sim}}(\vec{S}_{hc}) = \prod_{b=1}^{N_b} p_{\text{sim}}(\vec{s}_{hcb}).
\tag{21}
$$

We use the label "data" for the probabilities that describe the measured distributions and the equivalent expressions of eqs. (18) to (21), *i.e.*, $p_{\text{data}}(e_h)$, $p_{\text{data}}^{\text{rej}}$, $p_{\text{data}}^{\text{acc}}$ and $p_{\text{data}}(\vec{S}_{hc})$, respectively. With real measured data, these probabilities may only be approximate, but for the synthetic data that we use as an example in this paper, we know these exact probabilities must exist. Below, we describe how the best estimate for $p_{\text{data}}(\vec{s}_{hcb})$ is found.

For step 2, rather than directly working with $p_{\text{data}}(\vec{s}_{hcb})$ and $p_{\text{sim}}(\vec{s}_{hcb})$, we introduce weights by which the baseline simulation model results need to be reweighted in order to match the measured data. The exact weight for a single emission is given by, see also eq. (12),

$$
w_{\text{s}}^{\text{exact}}(\vec{s}_{hcb}) = \frac{p_{\text{data}}(\vec{s}_{hcb})}{p_{\text{sim}}(\vec{s}_{hcb})},
\tag{22}
$$

and the corresponding weight for the event $e_h$ is given by,[10]

$$
w_{\text{exact}}(e_h) = \frac{p_{\text{sim}}^{\text{acc}}}{p_{\text{data}}^{\text{acc}}} \left\langle w_{\text{exact}}(\vec{S}_{hN_h}) \right\rangle_{e(\vec{S}_{hN_h})=e_h},
\tag{23}
$$

---

[9]Note that the $N_h$ index for $\vec{S}_{hN_h}$ in eq. (18) is a dummy index, and thus the $p_{\text{rej}}$ is truly independent from the second term in eq. (18).

[10]In a Bayesian context, the weight in terms of $e$ is the evidence ratio between models obtained by marginalizing over all possible histories $h$ that are compatible with $e$.

with

$$w_{\text{exact}}(\vec{S}_{hN_h}) = \prod_{b=1}^{N_{\text{had}}-2} w_s^{\text{exact}}(\vec{s}_{hN_h b}), \tag{24}$$

where $N_{\text{had}}$ is the number of hadrons in the event. Here, the number of hadrons is two larger than the number of string breaks in the accepted simulation chain due to `finalTwo`, *i.e.*, in our notation $N_{\text{had}} - 2 = N_{h,N_h}$. In eq. (23), the product of single weights is averaged over fragmentation chains that produce the same event, see also eq. (18).

To achieve $w_{\text{exact}}(\vec{s}_{hcb}) \approx w_{\text{infer}}(\vec{s}_{hcb})$, several approximations can be made. First, it is unlikely to encounter two fragmentation chains simulated with very similar observable kinematics. In HOMER we can thus replace the average in eq. (23) with the weight for a single fragmentation chain[11]

$$w_{\text{infer}}(e_h, \theta) = \frac{p_{\text{sim}}^{\text{acc}}}{p_{\text{infer}}^{\text{acc}}(\theta)} w_{\text{infer}}(\vec{S}_{hN_h}, \theta), \tag{25}$$

where

$$w_{\text{infer}}(\vec{S}_{hN_h}, \theta) = \prod_{b=1}^{N_{\text{had}}-2} w_s^{\text{infer}}(\vec{s}_{hN_h b}, \theta). \tag{26}$$

To find the form of $w_{\text{infer}}(e_h, \theta)$, we parameterize $w_s^{\text{infer}}$ using a neural network $g$ with parameters $\theta$,

$$w_s^{\text{infer}}(\vec{s}_{hcb}, \theta) = g_\theta(\vec{s}_{hcb}). \tag{27}$$

The two acceptance probabilities in eq. (25) are therefore given by

$$p_{\text{sim}}^{\text{acc}} = \frac{N_{\text{acc}}}{N_{\text{tot}}}, \qquad p_{\text{infer}}^{\text{acc}}(\theta) = \frac{\sum_{\vec{S}_{jk} \in \{\vec{S}_{hc}^{\text{acc}}\}} w_{\text{infer}}(\vec{S}_{jk}, \theta)}{\sum_{\vec{S}_{jk} \in \{\vec{S}_{hc}\}} w_{\text{infer}}(\vec{S}_{jk}, \theta)}, \tag{28}$$

where $N_{\text{acc}} = N_{\text{sim}}$ is the number of accepted fragmentation chains in the simulation with $N_{\text{sim}}$ events, while $N_{\text{tot}}$ is the total number of chains in the fragmentation history, including the rejected ones.

The neural network of eq. (27) takes as input the seven-dimensional string break vector, $\vec{s}_{hcb}$ given by eq. (4), and outputs the weight $w_s^{\text{infer}}$ for this string break. Since the event weight, $w_{\text{infer}}(e_h, \theta)$, involves products of multiple weights, see eq. (25), it is easier to learn the logarithm of $w_s$. In fact, it is numerically expedient to introduce $\ln g_\theta(\vec{s})$ as a difference of two neutral networks

$$\ln g_\theta(\vec{s}) = g_1(z, \Delta\vec{p}_{\text{T}}, m, \text{fromPos}, \vec{p}_{\text{T}}^{\text{string}}; \theta) - g_2(\vec{p}_{\text{T}}^{\text{string}}; \theta), \tag{29}$$

where $\theta$ denotes the parameters of the neural network.

This choice of parameterization is not strictly necessary, however, it does allow us to impose the conditional structure eq. (9) explicitly in the loss function, see discussion surrounding eq. (36) below. The estimator for the fragmentation chain weight, $w_{\text{infer}}(\vec{S}_{hN_h}, \theta)$, is obtained by combining all individual $g_\theta$ contained in that chain, *c.f.* eq. (26). We do this by treating each chain $\vec{S}_{hc}$ as a string-break point cloud, not to be confused with the hadron-level point cloud discussed in section 2.2.1, and implementing eq. (29) as a module in a Message-Passing Graph Neural Network (MPGNN) written using the PYTORCH GEOMETRIC library [20].

The loss function for $g_\theta$ has two terms

$$\mathcal{L}_{\text{infer}} = \mathcal{L}_C + \mathcal{L}_{12}, \tag{30}$$

---

[11]Approximating eq. (23) with eq. (25), while accurate enough for the case of $q\bar{q}$ strings of fixed energy, was found empirically to break down when gluons are added to the string. A modified version of HOMER will therefore be needed in order to handle more general string hadronization cases [9].

where $\mathcal{L}_C$, given in eq. (34) below, ensures that the event-level weights $w_{\text{infer}}(e_h, \theta)$ reproduce the weights $w_{\text{class}}(e_h)$ of eq. (11) well, which were learned in step 1. The second term, $\mathcal{L}_{12}$, given in eq. (36) below, is a regularization term that ensures a proper convergence of the $g_1$ and $g_2$ NNs towards a solution that satisfies the conditional structure imposed by the string breaks within the Lund string model. In the remainder of this section, we motivate the forms of these two loss functions.

The main ingredient that makes step 2 of the HOMER method possible is that in step 1 we obtained a good approximation for the event weights, $w_{\text{class}}(e_h)$. All we need to ensure in step 2 is that $w_{\text{infer}}(e_h, g_\theta)$ reproduces well $w_{\text{class}}(e_h)$, and thus $w_{\text{exact}}(e_h)$, by minimizing the appropriate loss function. One possibility is to treat this as a regression problem and minimize an MSE loss

$$\mathcal{L}_R = \frac{1}{N_{\text{sim}}} \sum_{h=1}^{N_{\text{sim}}} (w_{\text{class}}(e_h) - w_{\text{infer}}(e_h, g_\theta))^2 . \tag{31}$$

However, for a finite dataset the MSE loss may not force $w_{\text{infer}}(e_h, g_\theta)$ to behave as a likelihood ratio. A conceptually clearer approach is to view the problem of constructing $w_{\text{infer}}(e_h, \theta)$ as yet another classification problem, where the learnable function observes both data and simulation to obtain the necessary likelihood-ratio. If we had access to histories for both simulation and measurements we could obtain the NN parameters in $g_\theta$ by minimizing the BCE loss

$$\mathcal{L}_C^{\text{BCE}} = -\mathbb{E}_{\text{sim}}\left[ \ln\left( \frac{1}{1 + w_{\text{infer}}(e_h, g_\theta)} \right) \right] - \mathbb{E}_{\text{data}}\left[ \ln\left( \frac{w_{\text{infer}}(e_h, g_\theta)}{1 + w_{\text{infer}}(e_h, g_\theta)} \right) \right], \tag{32}$$

where the two expectation values $\mathbb{E}$ are over simulation and data, respectively. The above loss function is minimized when $w_s^{\text{infer}}(\vec{s}_{hcb}) = p_{\text{data}}(\vec{s}_{hcb})/p_{\text{sim}}(\vec{s}_{hcb})$. While the fragmentation histories are not accessible in data, the expectation value over data in eq. (32) can be approximated through the use of the event weights that were learned in step 1. That is, for any observable $\mathcal{O}$ the expectation value over events is given by

$$\mathbb{E}_{\text{data}}[\mathcal{O}(e_h)] = \mathbb{E}_{\text{sim}}[w(e_h)\mathcal{O}(e_h)]. \tag{33}$$

This allows us to rewrite the BCE loss function as

$$\mathcal{L}_C = -\frac{1}{N_{\text{sim}}} \sum_{h=1}^{N_{\text{sim}}} \left( \ln\left( \frac{1}{1 + w_{\text{infer}}(e_h, g_\theta)} \right) + w_{\text{class}}(e_h) \ln\left( \frac{w_{\text{infer}}(e_h, g_\theta)}{1 + w_{\text{infer}}(e_h, g_\theta)} \right) \right), \tag{34}$$

which is minimized when $w_{\text{infer}}(e_h, g_\theta) = w_{\text{class}}(e_h)$. That is, we can achieve the same objective as the regression problem by taking the expectation value over simulated events, where we consider each event twice,[12] once unweighted and then once again weighted by $w_{\text{class}}$. This approach regularizes the problem by ensuring that the likelihood ratio behaves well both for simulated events and for measured events, where for the latter the distributions are approximated via the reweighted simulations.

The second term in the loss function, $\mathcal{L}_{12}$, treats $g_1$ and $g_2$ differently, and ensures that the individual string break weights, $w_s^{\text{infer}}(\vec{s}_{hcb}, \theta)$, are given by the ratios of conditional probabilities, see also eq. (22),

$$w_s^{\text{infer}}(\vec{s}_{hcb}, \theta) \approx w_s^{\text{exact}}(\vec{s}_{hcb}) = \frac{p_{\text{data}}\left( p(\{z, \Delta\vec{p}_{\text{T}}, m, \text{fromPos}\}_{h,c,b} | \{\vec{p}_{\text{T}}^{\text{ string}}\}_{h,c,b}) \right)}{p_{\text{sim}}\left( p(\{z, \Delta\vec{p}_{\text{T}}, m, \text{fromPos}\}_{h,c,b} | \{\vec{p}_{\text{T}}^{\text{ string}}\}_{h,c,b}) \right)} . \tag{35}$$

The form of the loss function that ensures this property is given by,

$$\mathcal{L}_{12} = -\sum_{\vec{s}_{hbc}} \left( \ln\left( \frac{1}{1 + \exp\left[ g_2(\vec{s}_{hbc}) \right]} \right) + \exp\left[ g_1(\vec{s}_{hbc}) \right] \ln\left( \frac{\exp\left[ g_2(\vec{s}_{hbc}) \right]}{1 + \exp\left[ g_2(\vec{s}_{hbc}) \right]} \right) \right). \tag{36}$$

---

[12]A feature of using the same data twice is that statistical fluctuations cancel out, see, *e.g.*, ref. [21].

In the limit of infinite baseline model simulation, *i.e.*, training data, $\mathcal{L}_{12}$ is minimized by $g_1$ and $g_2$ that satisfy

$$
\begin{aligned}
\exp\left[g_2(\vec{p}_{\mathrm{T}}^{\,\mathrm{string}})\right] = \int \mathrm{d}\Omega\, p_{\mathrm{sim}}(z, \Delta\vec{p}_{\mathrm{T}}, m, \mathtt{fromPos}|\vec{p}_{\mathrm{T}}^{\,\mathrm{string}}) \\
\times \exp\left[g_1(z, \Delta\vec{p}_{\mathrm{T}}, m, \mathtt{fromPos}, \vec{p}_{\mathrm{T}}^{\,\mathrm{string}})\right],
\end{aligned}
\tag{37}
$$

where $\Omega$ denotes the variables that are being sampled, *i.e.*, all the variables except the transverse momentum of the string, $\vec{p}_{\mathrm{T}}^{\,\mathrm{string}}$.

In the infinite simulation sample limit, the following relations hold,

$$
\exp\left[g_1(z, \Delta\vec{p}_{\mathrm{T}}, m, \mathtt{fromPos}, \vec{p}_{\mathrm{T}}^{\,\mathrm{string}})\right] \to \frac{p_{\mathrm{class}}(\{z, \Delta\vec{p}_{\mathrm{T}}, m, \mathtt{fromPos}, \vec{p}_{\mathrm{T}}^{\,\mathrm{string}}\})}{p_{\mathrm{sim}}(\{z, \Delta\vec{p}_{\mathrm{T}}, m, \mathtt{fromPos}, \vec{p}_{\mathrm{T}}^{\,\mathrm{string}}\})}\,,
\tag{38a}
$$

$$
\exp\left[g_2(\vec{p}_{\mathrm{T}}^{\,\mathrm{string}})\right] \to \frac{p_{\mathrm{class}}(\vec{p}_{\mathrm{T}}^{\,\mathrm{string}})}{p_{\mathrm{sim}}(\vec{p}_{\mathrm{T}}^{\,\mathrm{string}})}\,,
\tag{38b}
$$

$$
\exp\left(g_1 - g_2\right) \to w_{\mathrm{s}}\,,
\tag{38c}
$$

where $p_{\mathrm{class}}$ refers to the probability distributions obtained with $w_{\mathrm{s}}^{\mathrm{class}}$ such that we have $w_{\mathrm{infer}}(e_h, g_\theta) = w_{\mathrm{class}}(e_h)$ and $p_{\mathrm{class,sim}}(\vec{p}_{\mathrm{T}}^{\,\mathrm{string}})$ are the marginal distributions over the transverse momenta of the string. These are obtained by integrating out all the other variables. Equation (38c) is the limit of $\mathcal{L}_C$, and $\mathcal{L}_{12}$ enforces eq. (37) which produces the limits of eqs. (38a) and (38b). Combined, this results in the limiting behavior for $\mathcal{L}_{\mathrm{infer}}$, the loss function of eq. (30). We observe that in this infinite simulation sample limit, the parameterization of eq. (29) and the presence of the $\mathcal{L}_{12}$ term in $\mathcal{L}_{\mathrm{infer}}$ ensure that we are computing the weight between two conditional distributions, as dictated by how PYTHIA samples string breaks, avoiding other more expensive solutions, see *e.g.*, ref. [22].

### 2.2.3 Step 3: HOMER output

Once the weights for each individual string fragmentation $w_{\mathrm{s}}^{\mathrm{infer}}(\vec{s}_{hcb}, \theta)$ are known, it is straightforward to reweight any baseline simulation model fragmentation history. The output of HOMER is the weight, which is a product of the weights for all the string fragmentations in the baseline model simulation fragmentation history, including the rejected string fragmentations,

$$
w_{\mathrm{HOMER}}(\vec{\mathbf{S}}_h) = \prod_{c=1}^{N_h}\prod_{b=1}^{N_{hc}} w_{\mathrm{s}}^{\mathrm{infer}}(\vec{s}_{hcb}, \theta)\,,
\tag{39}
$$

where $N_{hc}$ are the string breaks contained in chain $c$. Compared to the event weight $w_{\mathrm{exact}}(e_h)$ of eq. (23), which contains averaging over histories that lead to the same event, the HOMER output is a weight for each individual fragmentation history. The event weight is the average of the compatible history weights. That is,

$$
w_{\mathrm{exact}}(e_h) \simeq w_{\mathrm{HOMER}}(e_h) \equiv \langle w_{\mathrm{HOMER}}(\vec{\mathbf{S}}_h)\rangle_{e(\vec{\mathbf{S}}_h)=e_h}\,.
\tag{40}
$$

Note that the event weight inferred in step 2, *i.e.*, $w_{\mathrm{infer}}(e_h)$ of eq. (25), differs from $w_{\mathrm{HOMER}}(\vec{\mathbf{S}}_h)$, because of averaging over rejected fragmentations. That is, $w_{\mathrm{infer}}(e_h)$ in eq. (25) contains the ratio $p_{\mathrm{sim}}^{\mathrm{acc}}/p_{\mathrm{infer}}^{\mathrm{acc}}(\theta)$, while $w_{\mathrm{HOMER}}(\vec{\mathbf{S}}_h)$ is a weight for a particular instance of a simulated fragmentation history. Once averaged over all fragmentation histories that produce the same event, the two weights $w_{\mathrm{infer}}(e_h)$ and $w_{\mathrm{HOMER}}(\vec{\mathbf{S}}_h)$ coincide. However, the weight $w_{\mathrm{HOMER}}(\vec{\mathbf{S}}_h)$ can be calculated from a single baseline simulation model fragmentation history, *i.e.*, for our

baseline simulation model this weight can be calculated directly from a single PYTHIA event. This is critical for any practical application of the method, where the correction can be applied on an event by event level for the baseline simulation model.

More importantly, as we have shown, the expectation values that enter the calculation of event weights can be estimated efficiently using simulated Monte Carlo samples where we have access to the simulated histories. This allows us to accurately estimate the new fragmentation function *without explicit access to the analytic form of the baseline fragmentation function*. That is, the new fragmentation function $f_{\text{data}}$ is implicitly defined through $w_s^{\text{infer}}(\vec{s}_{hcb}, \theta)$. The value of $f_{\text{data}}$ for a particular bin in the lightcone momentum fraction, $z \in [z_i, z_{i+1})$, and in the squared transverse mass, $m_T^2 \in [m_{T,j}^2, m_{T,j+1}^2)$, can be obtained by reweighting a sample of string breaks $\vec{s}_{hcb}$ that were simulated using the baseline model. Explicitly, we have

$$f_{\text{data}}(z_i, m_{T,j}^2) = \mathbb{E}_{\vec{s}_{hcb} \sim \text{sim}}\left[ w_s^{\text{infer}}(\vec{s}_{hcb}) \right]\Big|_{z \in [z_i, z_{i+1}), m_T^2 \in [m_{T,j}^2, m_{T,j+1}^2)}, \tag{41}$$

where the averaging of the weights $w_s^{\text{infer}}$ is performed only over the string breaks that correspond to the particular $(z, m_T^2)$ bin. In section 3 we also consider $\langle f(z) \rangle$, the fragmentation function averaged over all the sampled $m_T^2$ values, which is thus given by

$$\langle f_{\text{data}}(z_i) \rangle = \mathbb{E}_{\vec{s}_{hcb} \sim \text{sim}}\left[ w_s^{\text{infer}}(\vec{s}_{hcb}) \right]\Big|_{z \in [z_i, z_{i+1})}. \tag{42}$$

Note that in both of the above expressions the only requirement on the baseline fragmentation function is that it can be sampled.

We emphasize that we are exploiting the fact that, for simulated events, we have both the underlying fragmentation history, as produced by the model, and observable quantities that can be compared with measurements. If we learn how to reweight a given history to match the measured event distribution, we are effectively updating the fragmentation model.

### 2.3 Contrasting the HOMER method with GANs

To recapitulate, HOMER learns a data-driven fragmentation function $f_{\text{data}}(z)$ by estimating the likelihood ratio $w_s$ for each string break, eq. (22), which depends on the yet-to-be-determined $f_{\text{data}}(z)$ and the baseline simulation model fragmentation function $f_{\text{sim}}(z)$. The HOMER method divides the task of learning $f_{\text{data}}(z)$ into three steps. In step 1, event-level observables $\vec{x}_h = \vec{x}_h(e_h)$ are used to estimate the event-level weights $w_{\text{exact}}(e_h)$. In step 2, the event-level weights $w_{\text{exact}}(e_h)$ are used to train two neural networks $g_1$ and $g_2$. These two neural nets then provide $w_s$ for each string break such that the reproduced $w(e_h)$ best matches the target value, $w_{\text{exact}}(e_h)$. The string break weights are then used to reweight the baseline simulation model output, including rejected fragmentations, in step 3. The loss function that is used in the training of the $g_{1,2}$ networks is given in eqs. (30), (34) and (36), while fig. 2 shows the flowchart of the HOMER method.

Upon initial inspection, the HOMER method may appear similar in spirit to the adversarial strategies employed when training GANs [3, 4]. However, the optimization sequence in the HOMER method is significantly different. Rather than incorporating the weights $w(e_h)$ into a GAN-like loss function of the form

$$\mathcal{L}_{\text{GAN}} = -\sum_{\text{sim}} w(e_h) \ln(1 - D(\vec{x}_h)) - \sum_{\text{data}} \ln D(\vec{x}_{h'}), \tag{43}$$

and alternating between minimization through $D(\vec{x}_h)$ updates and maximization through $w(e)$ updates, in HOMER the event-level weights $w(e_h)$, obtained during step 1, are first frozen, and then the individual string break weights, $w_s^{\text{infer}}(\vec{s}_{hcb})$, are learned in step 2 by explicitly relating

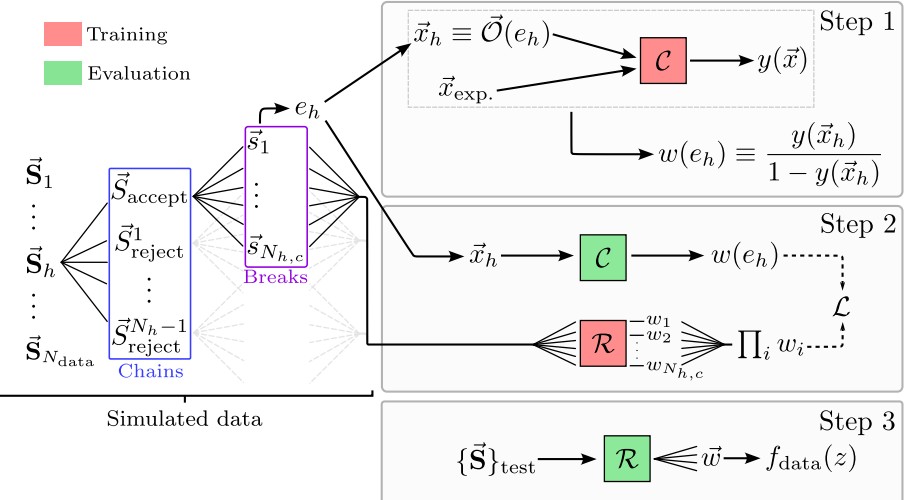

Figure 2: Summary flowchart of the HOMER method. Here $\mathcal{C}$ is the event-level classifier and $\mathcal{R}$ is the weight-level regressor that transforms the event-level weights into string break-level weights.

them to the event-level weights. The difference between the two approaches is more evident, if the BCE loss function $\mathcal{L}_C$ used in HOMER, eq. (34), is rewritten as

$$\mathcal{L}_C = -\frac{1}{N_{\text{sim}}} \sum_{\text{sim}} \left( \ln\left(\frac{1}{1+w(e_h)}\right) + \frac{D(\vec{x}_h)}{1-D(\vec{x}_h)} \ln\left(\frac{w(e_h)}{1+w(e_h)}\right) \right), \qquad (44)$$

where $D(\vec{x}_h)$ is obtained in step 1, and $w(e_h)$ is determined by the $g_{1,2}$ NNs in step 2.

Although both approaches require a very precise discriminator $D(\vec{x}_h)$, framing the training problem using the HOMER approach avoids certain pitfalls of an adversarial strategy at the expense of more involved computations. In particular, in the HOMER method, the estimate for the weight $w(e_h)$ is updated by minimizing a convex function. The solution does not correspond to an equilibrium between competing tasks but rather to a possible underlying model that yields the appropriate observables after integration over the unseen latent variables. While the HOMER method provides a regularized training and can be framed in fully probabilistic terms as an evidence ratio estimation, there is an added complication; in step 2 all the relevant details regarding the baseline simulation model need to be fully incorporated, *e.g.*, the PYTHIA `finalTwo` efficiency estimation, which can otherwise be ignored when trying to fool the discriminator.

# 3 Numerical results

To showcase the use of the HOMER method we used PYTHIA to generate two different sets of $2 \times 10^6$ events, in each case hadronizing a $u\bar{u}$ string with a center-of-mass frame energy of $\sqrt{s} = 90$ GeV, while permitting only emission of pions for simplicity. The two sets were generated using two different sets of values for the Lund parameters. The baseline simulation model dataset was generated using the default Monash values for the parameters, $a_{\text{sim}} = 0.68, b = 0.98, \sigma_{\text{T}} = 0.335$, while the synthetic measured data were generated for a changed value of $a_{\text{data}} = 0.30$, with all the other parameters kept the same. The choice of parameters was based on the benchmarks studied in ref. [10], where we found the reweighting from "sim" to "data" for this change in the Lund $a$ parameter to be non-trivial yet still achievable with good coverage, albeit with low effective statistics. Both the synthetic measured data

and baseline simulation model datasets were generated using PYTHIA, in order to be able to perform a closure test. In future applications, only the baseline simulation model dataset need be generated using PYTHIA, or another generator of choice, while the actual experimentally measured observables will be used for step 1.

The two $2 \times 10^6$ event datasets were both split in half, with $N_{\text{train}} = 10^6$ and $N_{\text{test}} = 10^6$ events in each dataset used for training and testing, respectively. All the figures below were obtained using the testing datasets, which were also used to verify the absence of any significant over-fitting both in step 1 and step 2 of the HOMER method. In more realistic applications, where HOMER is to be tuned to data instead of performing a closure test using simulated datasets, the datasets should be divided into three subsets for training, testing and visualization. This will avoid overfitting of the test dataset.

For step 1 we consider three different cases, distinguished by the level of available information. In section 3.1 only high-level observables were used to train the classifier in step 1. The high-level observables were either grouped into histograms, section 3.1.1, or were provided on an event-by-event basis, section 3.1.2. In section 3.2, a point-cloud representation of the events was used instead.

The training method of step 2 was same for all three cases. Only the input $w_{\text{class}}(e_h)$ weights from step 1 differ between these three cases, as detailed in sections 3.1 and 3.2. It is important to note that for all three cases, the string breaks in step 2 are described by seven-dimensional vectors $\vec{s}_{hcb}$, see eq. (4). These seven variables are further partitioned into two groups; the first group characterizes each string break, while the second group encodes the state of the string fragment before the break.

1. The five variables that are sampled by our baseline simulation model PYTHIA for a string break are: $\{z, \Delta p_x, \Delta p_y, m, \texttt{fromPos}\}$. Here, $z$ is sampled from the symmetric Lund string fragmentation function, eq. (3), $\Delta p_x$ and $\Delta p_y$ are sampled from a normal distribution of width $\sigma_{\text{T}}/\sqrt{2}$, and the hadron mass $m$ is randomly selected from either the mass of the $\pi^{\pm}$ or $\pi^0$ mesons. Finally, a binary random variable $\texttt{fromPos}$ determines whether the string break occurs on the negative or the positive end of the string. Inclusion of this variable is necessary to fully relate $z$ to $p_z$ and $E$, while also allowing to check that any learned hadronization function does not spoil the required left-right symmetry of the string.

2. The two remaining variables in $\vec{s}_{hcb}$ are the components of the transverse string momentum, $p_x^{\text{string}}$ and $p_y^{\text{string}}$, which describe the string state prior to the string break. The probability for a string break, $p_{\text{data}}(\vec{s}_{hcb})$, depends conditionally on these two variables, see eqs. (3) and (10).

To minimize the step 2 loss function, $\mathcal{L}_{\text{infer}}$ of eq. (30), we use an MPGNN implemented in the PYTORCH GEOMETRIC library [20]. We treat each fragmentation chain as a particle cloud with no edges between the nodes,[13] with string break vectors $\vec{s}_{hcb}$ with $b = 1, \ldots, N_{h,c}$, as the nodes. The learnable function $\ln g_\theta = g_1 - g_2$ corresponds to an edge function that is evaluated on each node and produces updated weights for that node. The updated weight $w_{\text{infer}}(\vec{S}_{hN_h}, \theta)$ for the whole fragmentation chain is then obtained by summing $\ln g_\theta$ over all the nodes and exponentiating the sum, *c.f.* eq. (26).

The $g_1$ and $g_2$ are fully connected neural networks with 3 layers of 64 neurons each and rectified-linear-unit, ReLU, activation functions. The inputs are either string break vectors $\vec{s}_{hcb}$ for $g_1$, or $\vec{p}_{\text{T}}^{\text{string}}$ for $g_2$, see eq. (29). The output of each neural network is a real number with no activation function applied. The weight for a single string break is given by

---

[13]We do not need to connect the string breaks since $\vec{s}_{hcb}$ already tracks the relevant information about the string state prior to the string break, $\vec{p}_{\text{T}}^{\text{string}}$, which enters the string break probability distributions, see eq. (3).

$w_s^{\text{infer}} = \exp(g_1 - g_2)$, *c.f.* eq. (27), which are then combined according to eq. (25) to produce the event weight, $w_{\text{infer}}(e_h)$. Note that the loss function to be optimized, $\mathcal{L}_{\text{infer}}$ in eq. (30), is a sum of two terms, the loss function $\mathcal{L}_C$ of eq. (34), which depends on event weights, and $\mathcal{L}_{12}$ of eq. (36), which depends directly on the values of $g_1$ and $g_2$ for each string break. We optimize $\mathcal{L}_{\text{infer}}$ using the Adam optimizer with an initial learning rate $10^{-3}$ that decreases by a factor of 10, if no improvement is found after 10 steps. We train for 100 epochs with batch sizes of $10^4$. To avoid over-fitting, we apply an early-stopping strategy with 20 step patience.

Next, we show the numerical results for the three different treatments of step 1, starting with high-level observables as inputs in section 3.1, while the results for point-cloud inputs are given in section 3.2.

## 3.1 High-level observables

The high-level observables that we use as inputs to the classifier in step 1 are the same 13 high-level observables that were used in the Monash tune [23], though now only for light flavors:

- Event shape observables: $1 - T$, $B_T$, $B_W$, $C$ and $D$; their definitions are collected in appendix B.

- Particle multiplicity $n_f$, the total number of visible particles in the event, and charged particle multiplicity $n_{\text{ch}}$, the number of charged particles in the event.

- The first three moments of the $|\ln x|$ distribution, the second and the third moment are computed around the mean, where $x$ is the momentum fraction of a particle. Explicitly $x = 2|\vec{p}|/\sqrt{s}$ where $\sqrt{s}$ is the center of mass of the collision and $\vec{p}$ the momentum of the particle. This is computed both for all visible particles, $\ln x_f$, and for just the charged particles, $\ln x_{\text{ch}}$.

These 13 high-level observables must all be calculated on an event-by-event basis. However, experimental measurements are currently only available for aggregate distributions of these observables, *i.e.*, histograms. Below, we thus distinguish two possibilities. In section 3.1.1 we first show results for the case where the classifier in step 1 is trained on binned distributions of high-level observables. In section 3.1.2 we then show by how much the performance of the HOMER method improves, if event-by-event information for high-level observables was available.

### 3.1.1 Using binned distributions in the step 1 classifier

We start the analysis by considering the case where the classifier in step 1 is trained on distributions of high-level observables. Such binned distributions are already available experimentally, see *e.g.*, ref. [23]. The results of this section can thus be viewed as a proxy for what can already now be achieved in determining the form of the Lund string fragmentation function from data.

The loss function that is being minimized in step 1 is given in eq. (16). We consider 10 bins for each continuous high-level variable, and natural binning for the $n_f$ and $n_{\text{ch}}$ multiplicities, *i.e.*, the bins are the values of the discrete variable, where for $n_{\text{ch}}$ only even values are allowed. The choice of a number of bins has a noticeable impact on the performance. This particular choice of 10 bins ensures enough statistics per bin, while avoiding the total number of bins from becoming so large that the curse of dimensionality enters. In the future, when existing measurements such as those considered in ref. [23] will be used, the binning will by necessity be dictated by the measured data. Additionally, correlations between observables can and should be included, when available.

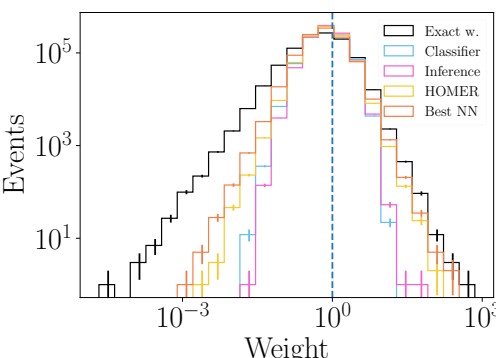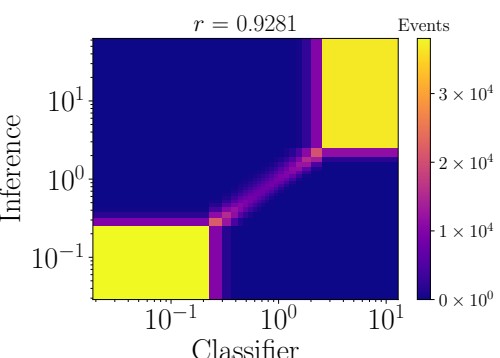

Figure 3: (left) The distributions of event weights $w(e_h)$ that follow from the HOMER method applied to the binned high-level observables, section 3.1.1. (right) Comparison between $w_{\text{class}}(e_h)$ from step 1, on the $x$ axis, and $w_{\text{infer}}(e_h, \theta)$ from step 2, on the $y$ axis. The closer the Pearson correlation coefficient $r$ is to 1, the closer the match between $w_{\text{class}}(e_h)$ and $w_{\text{infer}}(e_h, \theta)$, signaling better training during step 2.

The classifier in step 1 is a feed-forward NN implemented with the PYTORCH library [24]. To avoid over-fitting, we consider a small NN composed of two inner layers with 13 and 26 neurons each, and with ReLU activation functions. The final layer has a Sigmoid activation function to ensure $y(\vec{x}_h) \in [0, 1]$. Step 2 and step 3 are as described in sections 2.2.2 and 2.2.3, respectively. The results are shown in figs. 3 to 8 with the following labels:

- **Simulation**: The simulated distributions obtained using the baseline simulation model, *i.e.*, PYTHIA.

- **Data**: The experimentally measured distributions. In this work these are obtained from our synthetic data, produced from a PYTHIA simulation that uses the $a_{\text{data}}$ Lund parameter rather than $a_{\text{sim}}$. Close *Simulation* and *Data* distributions indicate that the baseline simulation model already describes the data well. The goal of HOMER is to reproduce the *Data* distributions.

- **HOMER:** The distributions that follow from reweighting the *Simulation* dataset using the per event weights $w_{\text{HOMER}}(e_h)$, *i.e.*, the outputs of the HOMER method given by eq. (39). A comparison between the *Data* and HOMER distributions is a gauge of the HOMER method fidelity.

We also give distributions for two intermediate results of the HOMER method, namely using the weights that were derived at the end of step 1 and step 2, respectively.

- **Classifier**: Here, the reweighting of the *Simulation* datasets was performed using the per event weights $w_{\text{class}}(e_h)$ from eq. (15), obtained as a result of the classifier training in step 1. A comparison of the *Data* and *Classifier* distributions is a gauge of the performance of the classifier used in step 1 of the HOMER method.

- **Inference**: Similar to *Classifier,* but using per event weights $w_{\text{infer}}(e_h, \theta)$ obtained in step 2. A comparison of *Inference* and HOMER distributions measures the difference in two ways of calculating the weights due to the rejected fragmentation chains, eq. (25) versus eq. (39). In the limit of an infinite baseline simulation sample size, the two should match for the observable distributions.



Figure 4: Distributions of high-level observables $1-T$, $C$, $D$, $B_W$ and $B_T$, for definitions see appendix B, for the case where step 1 of the HOMER method is performed on the binned high-level observables. See the main text for details.

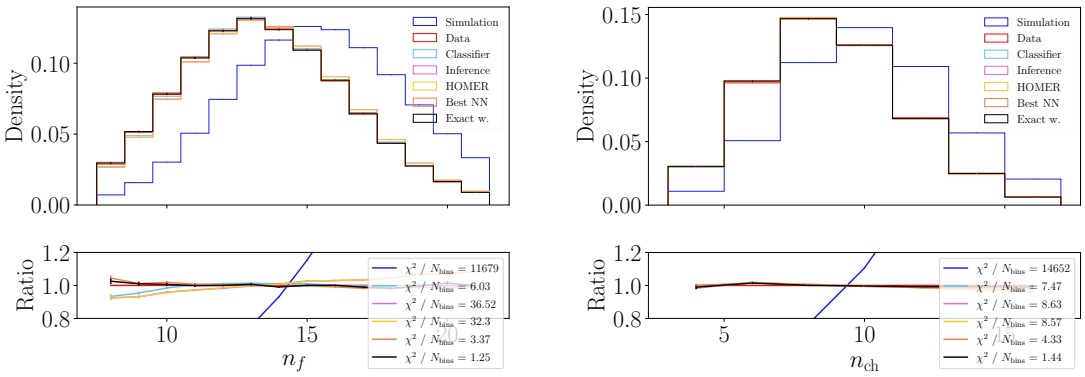

Figure 5: (left) The particle multiplicity $n_f$ and (right) the charged particle multiplicity $n_{\mathrm{ch}}$, for the case where step 1 of the HOMER method was performed using binned high-level observable distributions.

In addition, we show distributions that use the reweighting approach to transform *Simulation* distributions to *Data* distributions, but utilize information that is not available experimentally. In this way we can disentangle the inverse problem, how well we can learn $f(z)$ from data, from the statistical uncertainties that are introduced due to the use of reweighting and the use of NNs on finite samples. These two sets of distributions are labeled as follows.

- **Exact weights**: In this case the *Simulation* distributions are reweighted following ref. [10], including reweightings due to the rejected chains. The *Exact weights* and *Data* distributions should be nearly identical, except for increased statistical uncertainties in parts of the distributions due to differing supports of the underlying fragmentation functions, $f_{\mathrm{data}}(z)$ and $f_{\mathrm{sim}}(z)$. In this sense the *Exact weights* distributions represent an upper limit on the fidelity that can be achieved by the HOMER method, since this also uses reweighting.

- **Best NN**: In this case the $g_1$ and $g_2$ NNs are trained directly on exact single emission weights $w_s^{\mathrm{data}}(\vec{s}_{hbc})$, eq. (22), which are known only because both the synthetic data and baseline simulation model are from PYTHIA. The comparison of *Exact weights* and *Best NN* distributions is a measure of the $g_{1,2}$ NNs' fidelity. Since HOMER uses both reweighting and $g_{1,2}$ NNs, this is also an upper limit on the fidelity that can be achieved by the HOMER method, albeit a more realistically achievable one.

The HOMER, *Classifier*, *Inference*, *Exact weights*, and *Best NN* distributions of event weights $w(e_h)$ are shown in the left plot of fig. 3. We observe that the distributions of $w_{\mathrm{HOMER}}(e_h)$ match well both the *Best NN* and the *Exact weights* distributions, while being closer to the *Best NN* distribution, as expected, since this encodes both the approximations due to weighting and the use of $g_{1,2}$ NNs. The distribution of the weights for the intermediate results, the *Classifier* distribution from step 1 and *Inference* distribution from step 2, are further away, which is not surprising since they are truly the weights of the events, while the previous weights are calculated on individual histories.

In the right plot of fig. 3 we also show the comparison between step 1 *Classifier* and step 2 *Inference* event-level weights, $w_{\mathrm{class}}(e_h)$ and $w_{\mathrm{infer}}(e_h, \theta)$, respectively. Since the goal of step 2 is to obtain *Inference* weights that match the *Classifier* weights, their correlation is a possible metric of training success. The closer their correlation coefficient $r$ is to 1, the more successful the training. Deviations from identical reconstruction can be due to finite samples, imperfect optimization of the loss function, and the breakdown of the approximate expression for the

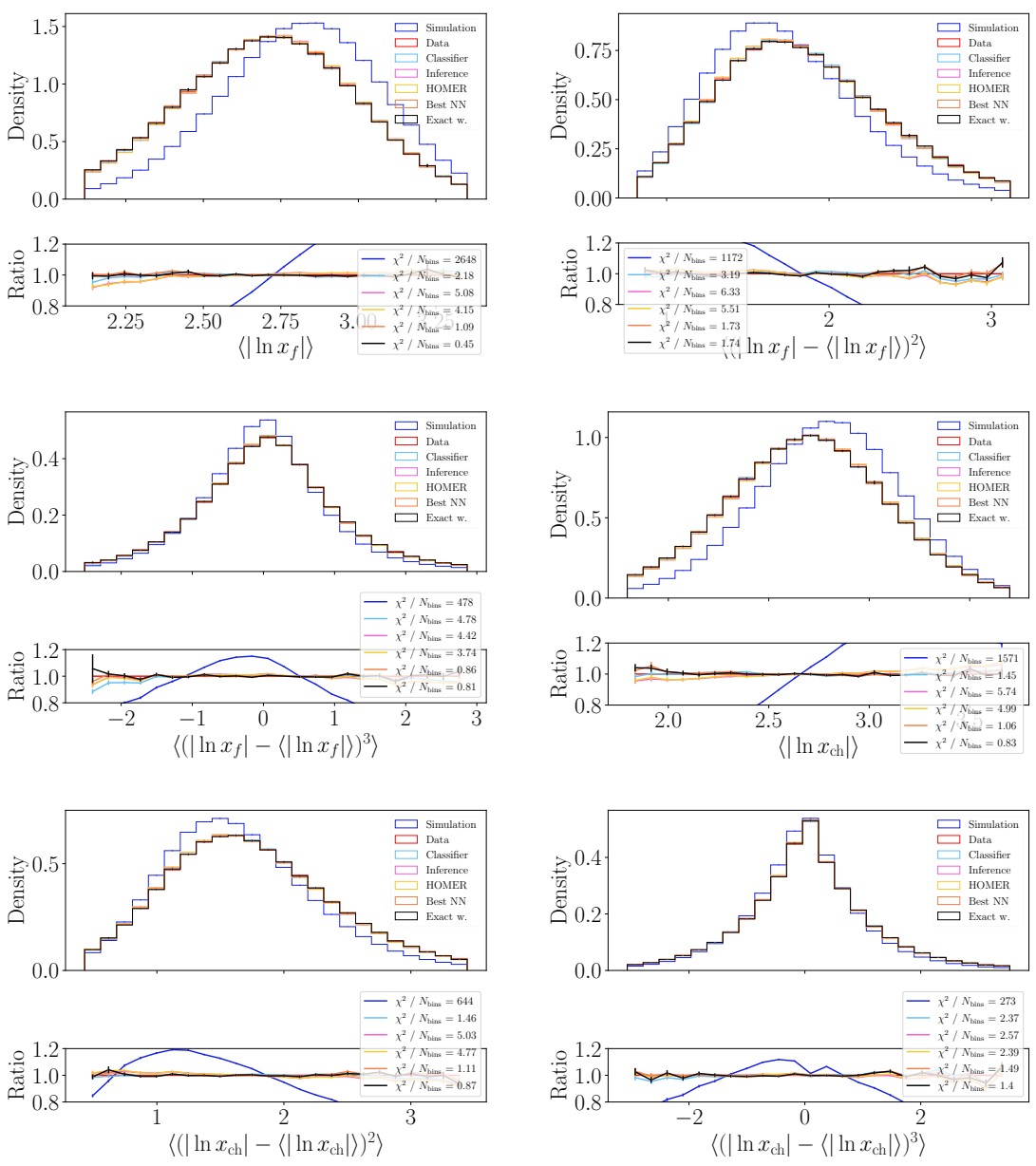

Figure 6: Distributions of the first three moments of $\ln x$, where $x = 2|\vec{p}|/\sqrt{s}$ for visible particle and charged particle distributions. Here, step 1 of the HOMER method is performed on binned high-level observables. See the main text for details.

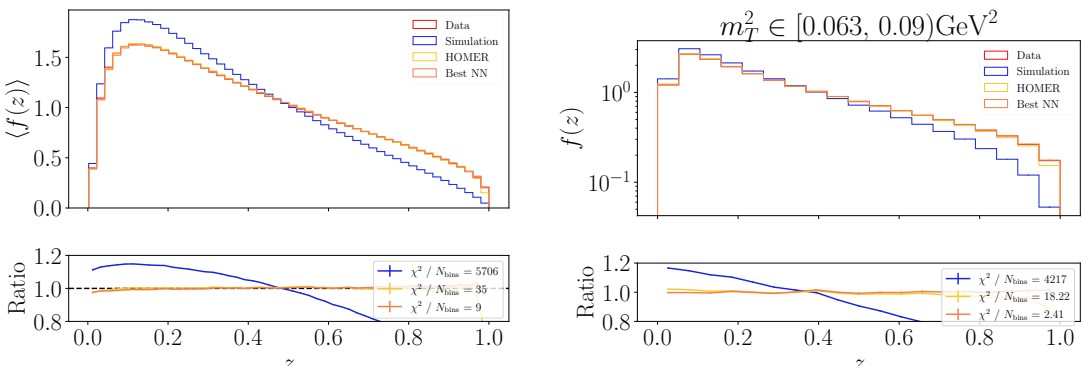

Figure 7: (left) Distributions for the fragmentation function averaged over all the string break variables except $z$, and (right) for when the transverse mass bin was kept fixed in addition. All model weights originate from the model trained with the binned high-level observables.

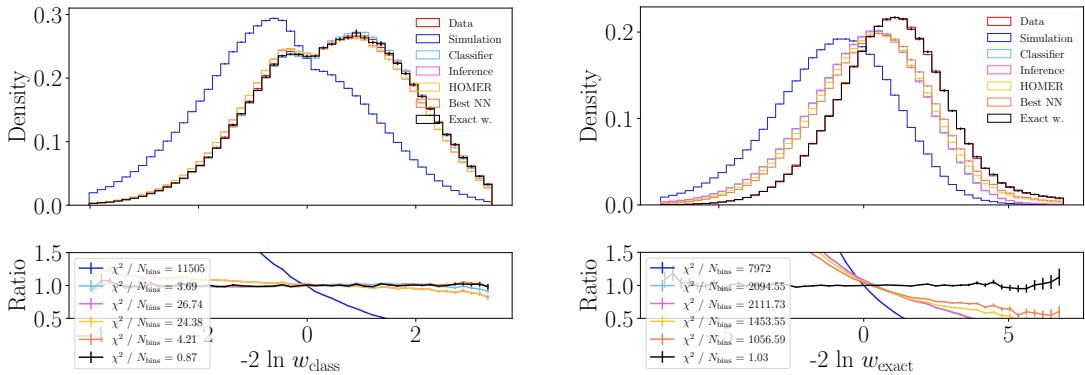

Figure 8: The distribution of the optimal observables (left) $-2\ln w_{\text{class}}$ and (right) $-2\ln w_{\text{exact}}$. The results are for a classifier trained on the binned high-level observables, see the text of section 3.1.1 for details.

*Inference* weight, eq. (25). The right plot results of fig. 3 show that a relatively successful training was achieved but is by no means perfect.

The distributions of high-level observables are shown in fig. 4 for $1-T$, $C$, $D$, $B_W$ and $B_T$, in fig. 5 for $n_f$ and $n_{\text{ch}}$, and in fig. 6 for the three moments of the $\ln x_f$ and $\ln x_{\text{ch}}$ distributions. To quantify the agreement between the reweighted simulated samples and the *Data* we also show the $\chi^2$ goodness of fit metric

$$\frac{\chi^2}{N_{\text{bins}}} = \frac{1}{N_{\text{bins}}} \sum_{k=1}^{N_{\text{bins}}} \frac{\left(p_{\text{data},k}^{\mathcal{O}} - p_{\text{pred},k}^{\mathcal{O}}\right)^2}{(\sigma_{\text{data},k}^{\mathcal{O}})^2 + (\sigma_{\text{pred},k}^{\mathcal{O}})^2}, \tag{45}$$

where observable $\mathcal{O}$ was binned into $N_{\text{bins}}$ bins. Similar to the notation of eq. (16), $p_{\text{data},k}^{\mathcal{O}}$ denotes the fraction of experimental events in bin $k$, while $p_{\text{pred},k}^{\mathcal{O}}$ is the corresponding predicted fraction. The statistical uncertainties, Poisson and reweighted Poisson, on experimental and predicted distributions are denoted as $\sigma_{\text{data},k}^{\mathcal{O}}$ and $\sigma_{\text{pred},k}^{\mathcal{O}}$, respectively. Both the measurement and the simulation uncertainties are used in the definition of the goodness-of-fit metric in order to adequately account for the impact of low statistics in some of the bins.

From the results shown in figs. 4 to 6, we see that all the reweighted distributions, including the HOMER output, approximate well the *Data* distributions. However, while the *Exact weights* and *Best NN* distributions, which use information not available in data, reproduce data almost perfectly, there is some degradation of fidelity for both the HOMER output, as well as for the two intermediate results of the HOMER method, the *Classifier* and *Inference* distributions. Here, the use of event weights that are the result of step 1, *i.e.*, the *Classifier* distributions, perform better than the other two. This shows that there is some, though small, drop in fidelity when going from step 1 to step 2, that is, once $w_{\text{class}}$ is expressed in terms of a learned fragmentation function. Because the learned function is not perfect, the performance degrades.

Figure 7 shows the symmetric Lund string fragmentation function that is extracted from data using the HOMER method. Since the new function is expressed implicitly through the string-break weights $w_{\text{s}}^{\text{infer}}(\vec{s}_{hcb})$, we can plot it by binning the $z$ values of the individual string breaks sampled with the baseline simulation model and reweighted with $w_{\text{s}}^{\text{infer}}(\vec{s}_{hcb})$ as detailed in section 2.2.3. The left plot shows the string fragmentation function averaged over all the sampled values of $m_{\text{T}}^2$, which we denote as $\langle f(z) \rangle$, while the right plot shows $f(z)$ for a particular transverse mass squared bin, $m_{\text{T}}^2 \in [0.063, 0.09)\,\text{GeV}^2$. We see that the HOMER method is able to extract both the correct $\langle f(z) \rangle$ and the form of $f(z)$ at a fixed value of $m_{\text{T}}^2$. Over most of the range of $z$, the difference between the true form of $f(z)$ and the one extracted using the HOMER method are below few percent level, and are comparable with the *Best NN* result. Recall that the HOMER method does not require a parametric form of the fragmentation function, but rather learns the functional form from data. The numerical results in fig. 7 can be viewed as the main result of this paper, and show that, at least in principle, it is possible to solve the inverse problem and learn the form of the Lund string fragmentation function from data.

Given sizable datasets, it is possible to distinguish the HOMER, *Best NN* and true $f(z)$ on a statistical basis, as indicated by the values $\chi^2/N_{\text{bins}}$ of fig. 7 that are still much larger than one. This statement is more clearly illustrated with one-dimensional summary statistics. Based on the Neyman-Pearson lemma [25], we can use the likelihood ratio to construct a summary statistic which condenses the information that distinguishes between two hypotheses. In analogy to the test statistic used for hypothesis tests, we compute $-2\ln\frac{\mathcal{L}_{\text{data}}}{\mathcal{L}_{\text{sim}}}$, where $\frac{\mathcal{L}_{\text{data}}}{\mathcal{L}_{\text{sim}}}$ is the likelihood ratio between data and simulation. The likelihood ratio is computed either in terms of observable quantities, which are approximated by $w_{\text{class}}$, or using $w_{\text{exact}}$ which we have access to in our synthetic data. The distribution of $-2\ln w_{\text{class}}$ in the left plot of fig. 8 indicates that there is a difference between the learned and the true fragmentation functions, but it is not too large. However, this difference is much more prominent in the distribution of $-2\ln w_{\text{exact}}$, see the right plot of fig. 8, which indicates that the observables used so are not sufficient to fully determine the form of $f(z)$. We thus also explore in the two subsequent sections the training of the step 1 classifier on unbinned high-level data and on point-cloud datasets, respectively.

It is also important to keep in mind that the above analysis includes only statistical uncertainties. Given that the extracted and true forms of $f(z)$ mostly match at the percent level, it is reasonable to expect that the proper inclusion of systematic uncertainties will be more important than the above differences, once real data is used. That is, it is reasonable to expect that in view of other sources of uncertainties, that the accuracy achieved above from the binned high-level observables will likely suffice in practice.

### 3.1.2 Using unbinned distributions in the step 1 classifier

Next, we explore possible gains, if more information beyond the binned distributions considered in the previous section, should be measured in the future. We consider two possibilities: in this section we first assume that the same set of high-level observables that we considered in section 3.1.1 will be measured in the future on an event-by-event basis. That is, for each event

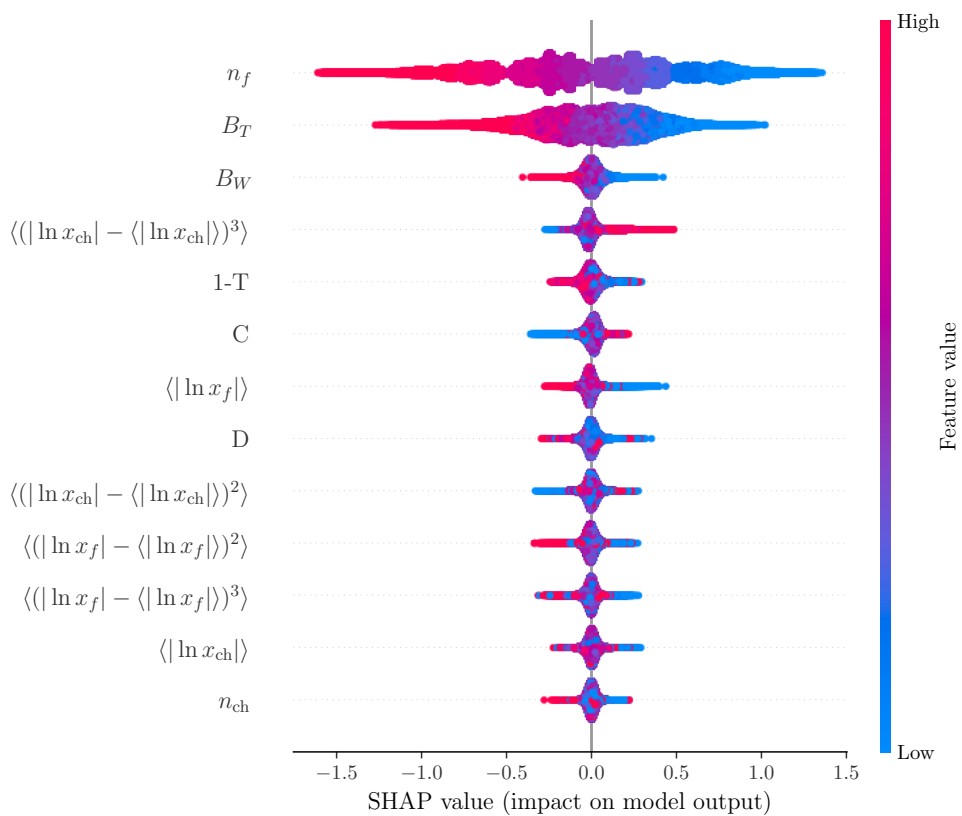

Figure 9: Shapley values for the classifier employed in step 1, using the unbinned high-level observables.

the 13 high-level observables would be measured, unlike in section 3.1.1, where only the distributions of these observables over entire experimental runs were assumed to be known. In section 3.2 we will then go one step further and assume that for each event the four momenta and the flavor labels for all of the outgoing hadrons are known. This will then represent the limit on information that can be extracted experimentally.

If we have access to the event-by-event information about the high-level observables, we can minimize the loss function of eq. (14) so that the outputs of the classifier then determines the event weights via eq. (15). For this task we used a gradient boosting classifier (GBC) implemented in the XGBOOST library [26], with a learning rate of 1, `lambda` of 0, `max_depth` of 10, `min_child_weight` of 1000, `colsample_bytree` of 0.5, `colsample_bylevel` of 0.5, `colsample_bynode` of 0.5, and all the other parameters set to their default values. The hyper-parameters were manually tuned such that the resulting classifier was smooth and well calibrated, since a well calibrated classifier is essential for successful training in step 2.

The use of a GBC rather than a feed forward NN is motivated by the GBC's speed, simplicity regarding hyper-parameter selection, and access to easily computable Shapley values [27, 28]. The Shapley values, shown in fig. 9 provide an estimate of the relative importance of each individual observable for the classifier. The Shapley values in fig. 9 show the multiplicity $n_f$ to be the most important feature, closely followed by $B_T$. Since multiplicity $n_f$ is correlated with charged multiplicity $n_{\text{ch}}$, while $B_T$ is correlated with the other event shape observables, fig. 9 implies that for determining the string fragmentation function the overall hadron multiplicity is more important than the event shape observables, at least in this particular scenario.

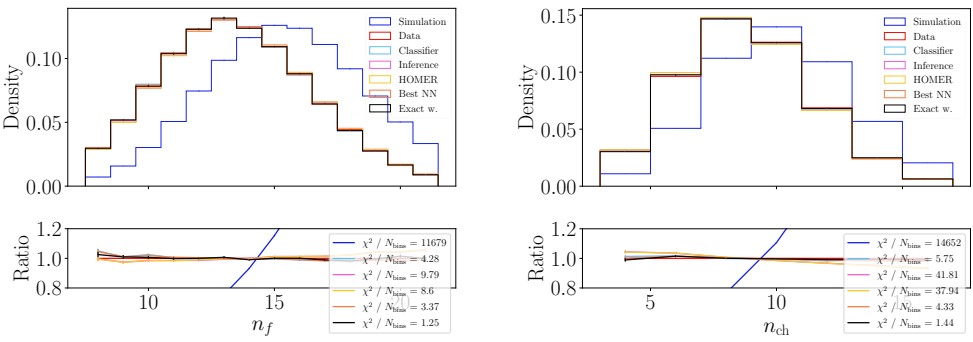

Figure 10: Distributions for the $n_f$ and $n_{ch}$ high-level observables; the distributions for other high-level observables are shown in appendix C.1. All weights are from the model trained with the unbinned high-level observables.

There are also correlation effects between $n_f$ and $B_T$ that are not captured by fig. 9. However, since the multiplicities are inherently infrared and collinear unsafe, we do expect them to always be sensitive to hadronization effects and have an important role, as reported in refs. [23, 29].

The improvement in the event-level weights compared to the case of binned distributions for high-level observables, section 3.1.1, is clearly visible in fig. 10, where we show the predictions for hadron and charged hadron multiplicities. The results for the other high-level observables are collected in appendix C.1, and show a similar trend. The $n_f$ and $n_{ch}$ *Classifier* distributions, obtained using the $w_{class}$ event weights from the step 1 classifier, are closer to the *Data* distribution, signaling better learning using event-by-event information, as expected. For hadron multiplicity, this improvement translates to a better performance for step 2 results, the *Inference* distributions from $w_{infer}$ weights, as well as for the final HOMER predictions. For charged multiplicity, the improvement is not present due to step 2 not perfectly reconstructing step 1.

The $\chi^2/N_{bins}$ for the HOMER distributions are even slightly smaller than for the *Inference* ones. This can be explained by the fact that the HOMER distributions are based on an instance of a full fragmentation history for an event, while the *Inference* ones use average acceptance probabilities for the rejected fragmentation chains. As can be seen in fig. 3, the HOMER distributions have a larger tail than the *Inference* ones, since the former are obtained by multiplying, on average, a larger number of individual string break weights. This larger variance then translates to a lower $\chi^2/N_{bins}$ for similar central values. Compared to the case where the step 1 classifier was trained on binned data, the $\chi^2/N_{bins}$ changed from about 32 (9) to 9 (38) for $n_f$ ($n_{ch}$), *c.f.* fig. 5. This is typical for all other high-level observables; for most, the $\chi^2/N_{bins}$ reduces significantly between the binned and unbinned case, unless the agreement with data was already reasonable.

This improvement in fidelity is perhaps best illustrated by the improved agreement between the extracted and true $f(z)$, in which case for $\langle f(z) \rangle$ the $\chi^2/N_{bins}$ drops from 35 to 7, *c.f.* figs. 7 and 11. Note that the $\chi^2/N_{bins}$ for HOMER extraction of $f(z)$ is very close to the *Best NN* one; the fragmentation function is almost as good as it can be. However, it still is not identical, *i.e.*, despite sub-percent agreement between the HOMER extracted and true $f(z)$, due to the large statistics available and no systematic uncertainties, we still have $\chi^2/N_{bins} \gg 1$. When extracting $f(z)$ from real data, the result could be improved by imposing more structure in the data-driven fragmentation function, in terms of specific functional dependencies on the different variables contained in $\vec{s}_{hcb}$. However, this risks turning HOMER into merely a more convoluted Lund string model.

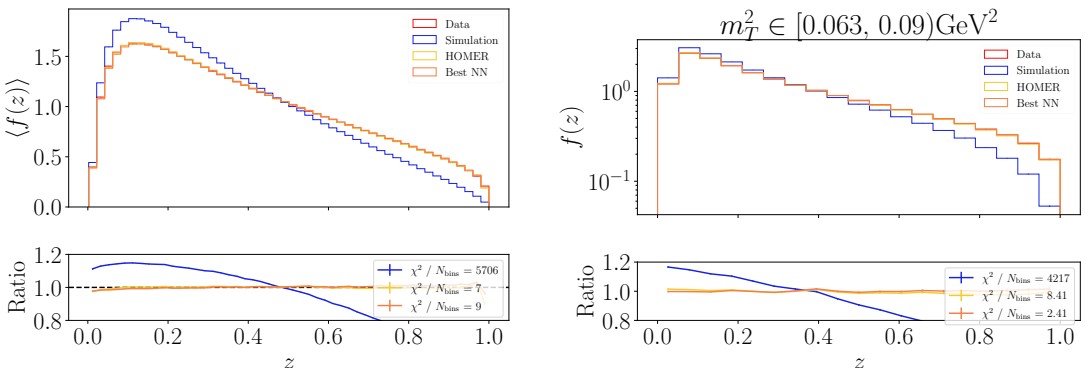

Figure 11: Distributions for the fragmentation function averaged over all string break variables except (left) $z$ and (right) fixing the transverse mass bin. All model weights originate from the model trained with the unbinned high-level observables.

## 3.2 Using point cloud in the step 1 classifier

Finally, we examine the use of the HOMER method with a point cloud representation of the event in the step 1 classifier. While in principle the point cloud contains all the available information about the event, including all the high-level observables studied in section 3.1, it is not clear *a priori* that this is the best form of the data to achieve a high fidelity extraction of $f(z)$ with limited resources. That is, in practice any algorithm trained on a point cloud has the added cost of extracting more useful representations for the task at hand. This cost can be expressed in the number of training samples needed.

In the following, we show how the HOMER method performs when the loss function of eq. (14) is minimized using a point cloud representation of the data. The $\vec{x}_h$ inputs to the classifier of eq. (14) are now unordered lists of particles as in $e_h$, eq. (8), containing the laboratory frame four momenta of hadrons. Thus, an event with $N_{\text{had}}$ hadrons is represented by an array of dimension $\{N_{\text{had}}, 4\}$. In practice, events are zero-padded for storage to a size of $\{100, 4\}$ and masking is used to restore the point cloud representation with variable multiplicity when training and evaluating the classifier.

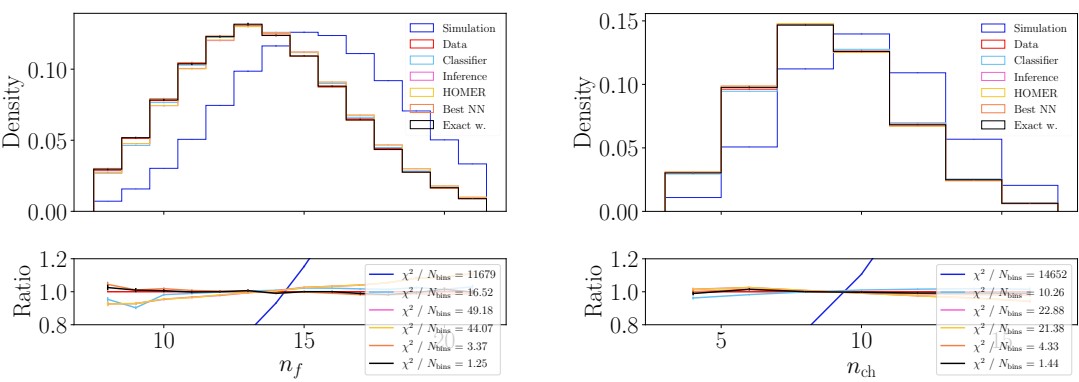

Figure 12: Distributions for the $n_f$ and $n_{\text{ch}}$ high-level observables; the distributions for other high-level observables are shown in appendix C.2. All weights are from the model trained with the with the point cloud representation.

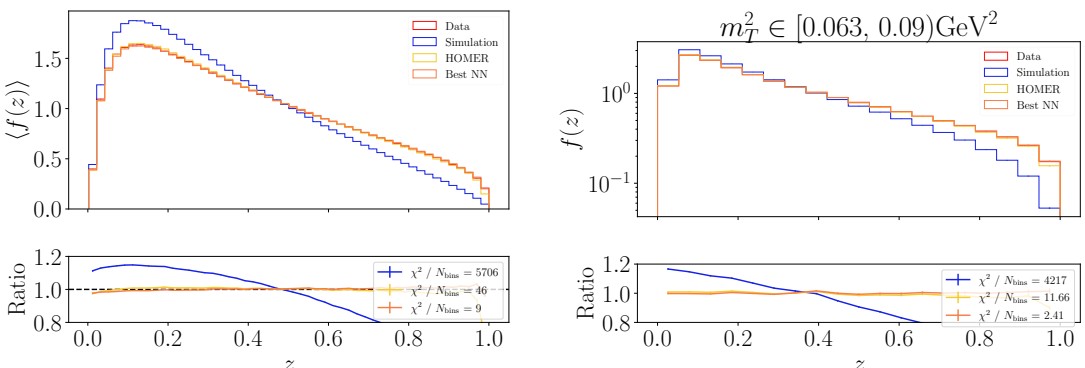

Figure 13: Distributions for the fragmentation function averaged over all string break variables except (left) $z$ and (right) fixing the transverse mass bin. All model weights originate from the model trained with the point cloud representation.

Our implementation of the point-cloud classifier is inspired by the PARTICLENET method [30] which uses a graph neural network to tag jets represented as particle clouds or Deep Sets, and is implemented in PYTORCH GEOMETRIC [20]. It consists of an MPGNN with two internal edge convolution layers. The edge convolution layers update each node by summing the edge function evaluated over all the neighbors of that node. To cluster the hadrons, we use $k$-nearest neighbors with $k = 8$. The two edge functions are neural networks with one inner layer of 64 neurons and a ReLU activation function. The first edge convolution returns a 64-dimensional embedding that is fed into the second-edge convolution, which again produces a 64-dimensional embedding. The point cloud is then summed over nodes to obtain a 64-dimensional vector which is fed to a multi-layer perceptron classifier with one inner layer of 64 neurons and a ReLU activation function, and an output layer consisting of one node and a Sigmoid activation function. The output of the classifier $y(\vec{x}_h) \in [0,1]$ is fed into the BCE loss function of eq. (14).

The predictions for $n_f$ and $n_{ch}$ are shown in fig. 12; additional results are collected in appendix C.2. While there is a degraded performance compared to the results based on high-level observables, shown in section 3.1, in the point-cloud case these high-level observables were never directly seen during training. The degraded performance is seen across all three steps of the HOMER method. The imperfect reconstruction of the weights from the step 1 classifier translates to a degraded performance in the predictions based on inference weights from step 2 and the final HOMER results. Ultimately, this is due to the increased inductive bias in the reconstructed fragmentation function, shown in fig. 13. However, while the learned fragmentation function is sub-optimal compared to the event-by-event high-level case, the difference between reconstructed and true $f(z)$ is still at the percent or even sub-percent level.

It will be interesting to see how these results carry over to an analysis based on real experimental data, which will have the added complication of experimental uncertainties due to detector and reconstruction effects. Nevertheless, a primary takeaway of our point-cloud analysis appears to be that full individual particle information does not provide significantly more information than event-by-event high-level observables, when learning hadronization models applied to simplified $q\bar{q}$ systems.

# 4 Conclusions and outlook

We have introduced the HOMER method for learning the fragmentation function $f(z)$ directly from data. The HOMER method reweights the probabilities for single hadron emissions in a baseline simulation model, such that the experimentally accessible observables match measurements. The HOMER method consists of three steps. In step 1 the classifier is trained on experimental data. This then provides an estimate of a probability weight by which each event of the baseline simulation model should be reweighted to match the experimental data. In step 2 this is then converted to an ML-based reweighting of single hadron emissions that are afterward combined into a weight for the full fragmentation history in the baseline simulation model.

As a proof of concept, we applied the HOMER method to a simple $q\bar{q}$ scenario where the relationship between fragmentation chains and observed events is rather straightforward. We studied three distinct cases for the measured observables which differ by the level of information available. The first case is binned distributions of high-level observables such as the particle multiplicities, thrust, *etc.*, all of which have already been experimentally measured. Thus, with this choice of observables the HOMER method could be immediately applied to existing experimental data. In the second case we assumed that the same high-level observables were measured on event-by-event basis. In the third and final case, the experimental data was assumed to be in the form of a point cloud, which in principle carries all the experimentally available information.

In all three cases, the HOMER method results in a fragmentation function $f(z)$ that is a very good approximation of the true $f(z)$ with which our synthetic measured data was generated; the difference between the true and learned $f(z)$ is at the percent level or below. While there is some degradation moving from unbinned to binned high-level observables, the achieved precision is already well below the anticipated experimental systematic uncertainties. This implies that for the simplified case of $q\bar{q}$ string fragmentation with fixed initial state kinematics, there is little incentive to perform measurements of high-level observables on an event-by-event basis. However, this is not necessarily true, and in fact we expect it not to be so, for the more complicated and realistic scenario of strings composed out of both quarks and gluons [9], as well as for complex topologies generated by color reconnection. The latter introduces further complications similar to the `finalTwo` issue, such as the convergence criteria in junction fragmentation [31, 32]. Additionally, the difference in performance when training HOMER using binned or unbinned high-level observables might increase significantly when the training is performed on actual measurements rather than performing a closure test on synthetic data. Further exploration in this direction, including comparison with dedicated event-by-event measurements, if these become available, is needed. Another interesting outcome of our proof-of-concept is that learning $f(z)$ from the unbinned high-level observables using the HOMER method already saturates the best possible NN-based description. Moving from unbinned high-level observables to a point-cloud trained classifier in step 1, we even find a slight degradation in performance, most likely due to the added difficulty in training.

In summary, we have demonstrated that fragmentation functions can be learned from data using the HOMER method. To make contact with real experimental data several extensions of presented results are required. First, the HOMER method should be extended to the case of strings with any number of attached gluons, as well as full flavor structure. Because we implemented the HOMER method using PYTHIA as the baseline simulation model, we expect our framework to translate straightforwardly to these types of extensions as they are already present in PYTHIA [9]. Second, a much larger change to the HOMER method would be to avoid the main underlying assumption of the method as presented in this manuscript, namely that the the Lund string fragmentation model sufficiently describes hadronization. At least in

principle, the method can be expanded to replace this specific choice of fragmentation model in its entirety, simply producing hadron collections from color singlets. To do this, state-of-the-art algorithms for point cloud generative models, *e.g.*, refs. [33–38], may prove useful. We leave such explorations for future work.

# Acknowledgments

We thank B. Nachman, T. Sjöstrand and P. Skands for their careful reading and constructive comments on the manuscript.

**Funding information** AY, JZ, MS, MW, PI, SM, and TM acknowledge support in part by NSF grants OAC-2103889, OAC-2411215 and OAC-2417682. AY, JZ, MS, and TM are also in part funded by DOE grant DE-SC101977. JZ acknowledges support in part by the Miller Institute for Basic Research in Science, University of California Berkeley. AY acknowledges support in part by The University of Cincinnati URC Graduate Support Program. SM is supported by the Fermi Research Alliance, LLC under Contract No. DE-AC02- 07CH11359 with the U.S. Department of Energy, Office of Science, Office of High Energy Physics. CB acknowledges support from Vetenskapsrådet contracts 2016-05996 and 2023-04316. MW and PI are also supported by NSF grant NSF-PHY-2209769. This work is supported by the Visiting Scholars Award Program of the Universities Research Association. This work was performed in part at Aspen Center for Physics, which is supported by NSF grant NSF-PHY-2210452.

# A Public code

The public code for this work can be found at https://gitlab.com/uchep/mlhad in the `HOMER/` subdirectory. The repository consists of a hierarchical structure with four major components: one for data generation and the other three corresponding to the three HOMER steps. Detailed explanations and instructions of usage can be found within the code documentation. All code is written in `Python` v`3.10.8`, and heavily utilizes the XGBOOST library version `1.7.6`, the PYTORCH library v`2.1.2` and the PYTORCH GEOMETRIC library v`2.4.0`. Finally, all datasets were produced using PYTHIA v`8.311`.

# B Definitions of shape observables

In this appendix we collect the definitions of shape observables $1 - T$, $B_T$, $B_W$, $C$ and $D$ that were used in section 3.1. Here, thrust is defined as [39, 40]

$$T = \frac{\sum_i |\vec{p}_i \cdot \vec{n}_T|}{\sum_i |\vec{p}_i|},$$
(B.1)

where the sum is over particles $i$ with three momenta $\vec{p}_i$, and the unit vector $n_T$ is chosen such that the above expression is maximized. The thrust takes values between 0.5 for spherical events and 1.0 for 2-jet events with narrow jets. The thrust axis $\vec{n}_T$ divides the space into two hemispheres, $S_\pm$, which are then used in the definitions of other shape variable.

The two jet broadening variables are obtained by computing for each hemisphere [41, 42]

$$B_\pm = \frac{\sum_{i \in S_\pm} |\vec{p}_i \times \vec{n}_T|}{2 \sum_i |\vec{p}_i|}.$$
(B.2)

The total jet broadening is then defined as

$$B_T = B_+ + B_-\,,\tag{B.3}$$

while the wide jet broadening is defined as

$$B_W = \max(B_+, B_-)\,.\tag{B.4}$$

The $C$ and $D$ parameters, on the other hand, are related to the eigenvalues of the linearized momentum tensor [43, 44]

$$\Theta^{ij} = \frac{1}{\sum_a |\vec{p}_a|} \sum_a \frac{p_a^i p_a^j}{|\vec{p}_a|}\,, \qquad i, j = 1, 2, 3\,,\tag{B.5}$$

where the summation is over different particles, with three momenta $\vec{p}_a$, while $p_a^i$ denotes component $i$ of the momentum. The three eigenvalues of $\Theta^{ij}$ are denoted $\lambda_{1,2,3}$, and the $C$ and $D$ parameters are

$$C = 3(\lambda_1\lambda_2 + \lambda_2\lambda_3 + \lambda_3\lambda_1)\,, \qquad D = 27\lambda_1\lambda_2\lambda_3\,.\tag{B.6}$$

## C  Additional numerical results

In this appendix we collect additional figures that supplement the results of section 3.1.2 for a step 1 classifier trained on unbinned high-level data in appendix C.1, and the results of section 3.2 for a step 1 classifier trained with a point cloud in appendix C.2.

### C.1  Step 1 with unbinned high-level observables

In this appendix we collect figures that supplement the ones shown in section 3.1.2, where the results of training the step 1 classifier using unbinned high-level observables was shown. The results of figs. 14 to 17, along with figs. 10 and 11 in the main text, mirror the results in figs. 3 to 8 of section 3.1.1, which were obtained using binned high-level observables.

Figure 15 shows distributions for the high-level observables $1-T$, $C$, $D$, $B_W$, and $B_T$, while fig. 16 shows the distributions for moments of $\ln x_f$ and $\ln x_{\mathrm{ch}}$. These are to be compared with the results of figs. 4 and 6 in section 3.1.1 of the main text, respectively. There is a significant reduction in $\chi^2/N_{\mathrm{bins}}$ for the shape observables when moving from binned to unbinned training, and similarly for the moments of $\ln x_f$ and $\ln x_{\mathrm{ch}}$, unless these were already captured well by the binned training.

Figure 14 shows the event weights that are obtained at different stages of the HOMER method, for the case of training on the unbinned high-level observables. The right plot in fig. 14 shows a comparison between the step 1 and step 2 event-level weights, $w_{\mathrm{class}}(e_h)$ and $w_{\mathrm{infer}}(e_h, \theta)$. We observe how each $w_{\mathrm{infer}}(e_h, \theta)$ is a satisfactory reconstruction of $w_{\mathrm{infer}}(e_h)$, although with a somewhat lower correlation coefficient $r$ than in the case of training on binned observables, *c.f.* fig. 3 of section 3.1.1. However, this slight decrease in correlation is less a reflection of worse performance than a consequence of a better underlying model that corrects any issues step 1 had by means of added inductive bias. As shown in the left figure, now the HOMER distribution is closer to the *Best NN* and thus a better approximation of the *Exact weights* distribution.

The lack of exactness is clearly evident in the optimal summary statistic shown in the right plot of fig. 17. Although the fragmentation function is a very good approximation, any differences accumulate quickly when computing the full history weight, and the optimal summary statistic reflects this. However, the main takeaway here is that HOMER is able to learn a good if not perfect approximation for the high-level observables, as evidenced by the summary statistic shown in the left plot of fig. 17.

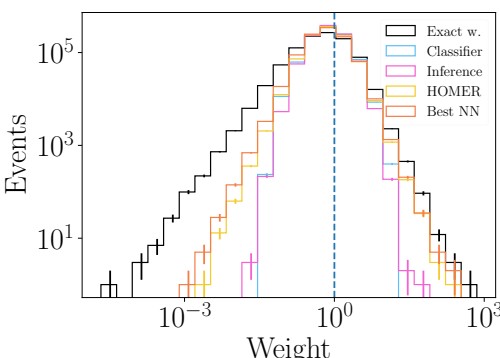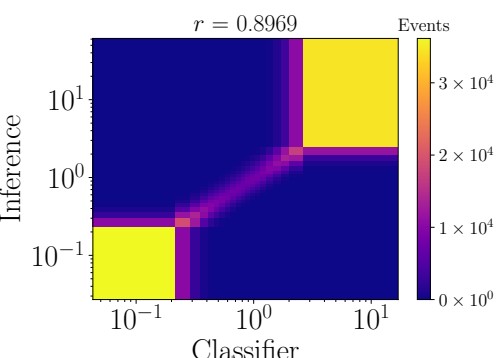

Figure 14: (left) The distributions of event weights $w(e_h)$ that follow from the HOMER method applied to the unbinned high-level observables, section 3.1.2. (right) Comparison between $w_{\text{class}}(e_h)$ from step 1, on the $x$ axis, and $w_{\text{infer}}(e_h, \theta)$ from step 2, on the $y$ axis. The closer the Pearson correlation coefficient $r$ is to 1, the closer the match between $w_{\text{class}}(e_h)$ and $w_{\text{infer}}(e_h, \theta)$, signaling better training during step 2.

## C.2 Point cloud case

In this appendix we collect figures that supplement the ones shown in section 3.2. The event weights are shown in fig. 18. The results show that the weights obtained in step 2 are a very good reconstruction of the weights from step 1, though the distribution of the HOMER weights is still not as good of an approximation of the *Exact weights* as the *Best NN* weights are. There are several possible sources of this slight degradation in performance. First, it could be due to the additional cost of extracting useful information from the point cloud representation of data, compared to the data projected on a set of high-level observables, resulting in reduced performance compared with the high-level observables. However, it could also be due to the relative simplicity of the $q\bar{q}$ string fragmentation example, for which, as shown in section 3.1.2, multiplicity is the main feature that distinguishes between the measured data and the baseline simulation model.

The difference in performance is consistently reflected in the distributions of the high-level observables. This is not surprising, since unlike the case of the step 1 classifier trained on either binned or unbinned data, here, the high-level observables are not seen directly during the training. The resulting distributions for the point cloud case are shown in figs. 19 and 20, see also fig. 12 in the main text. These are to be compared with figs. 15 and 16 in appendix C.1, and with fig. 10 in the main text, respectively.

We see that the results of step 1, the *Classifier* distributions, show a slightly degraded performance compared to the case where the classifier is trained directly on the high-level distributions. The step 2 *Inference* and the HOMER distributions similarly show an imperfect reconstruction of step 1. This degrades the reconstruction performance due to the slight increase in the inductive bias, also in the reconstruction of the fragmentation function, shown in fig. 13. That is, the $\langle f(z) \rangle$ reconstructed using the point cloud, shown in fig. 13, has a $\chi^2/N_{\text{bins}} = 46$, to be compared with a significantly smaller $\chi^2/N_{\text{bins}} = 9$ for the case of unbinned high-level observables, *c.f.* . fig. 11. Though, it is worth reiterating that the differences between learned and true $\langle f(z) \rangle$ are in both cases at or below percent level, and thus more than suffice in practice.

The summary statistics of fig. 21 also capture the degradation in performance of the point cloud case compared to the unbinned high-level observables one. Signs of sub-optimal training due to the added difficulties of training a point cloud can be found in the results shown

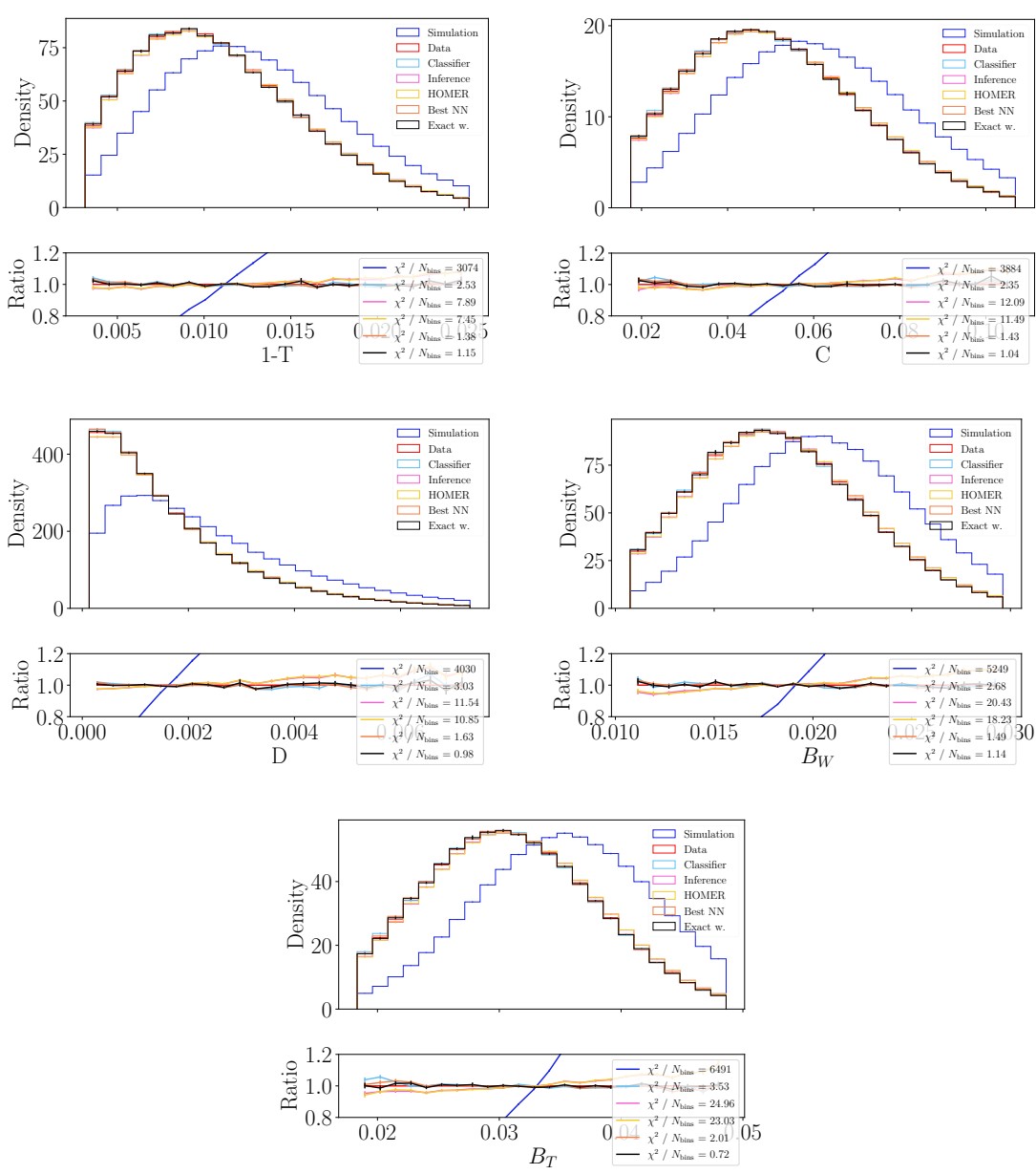

Figure 15: Distributions of high-level observables $1 - T$, $C$, $D$, $B_W$ and $B_T$, for definitions see appendix B, for the case where step 1 of the HOMER method is performed on the unbinned high-level observables. See the main text for details.

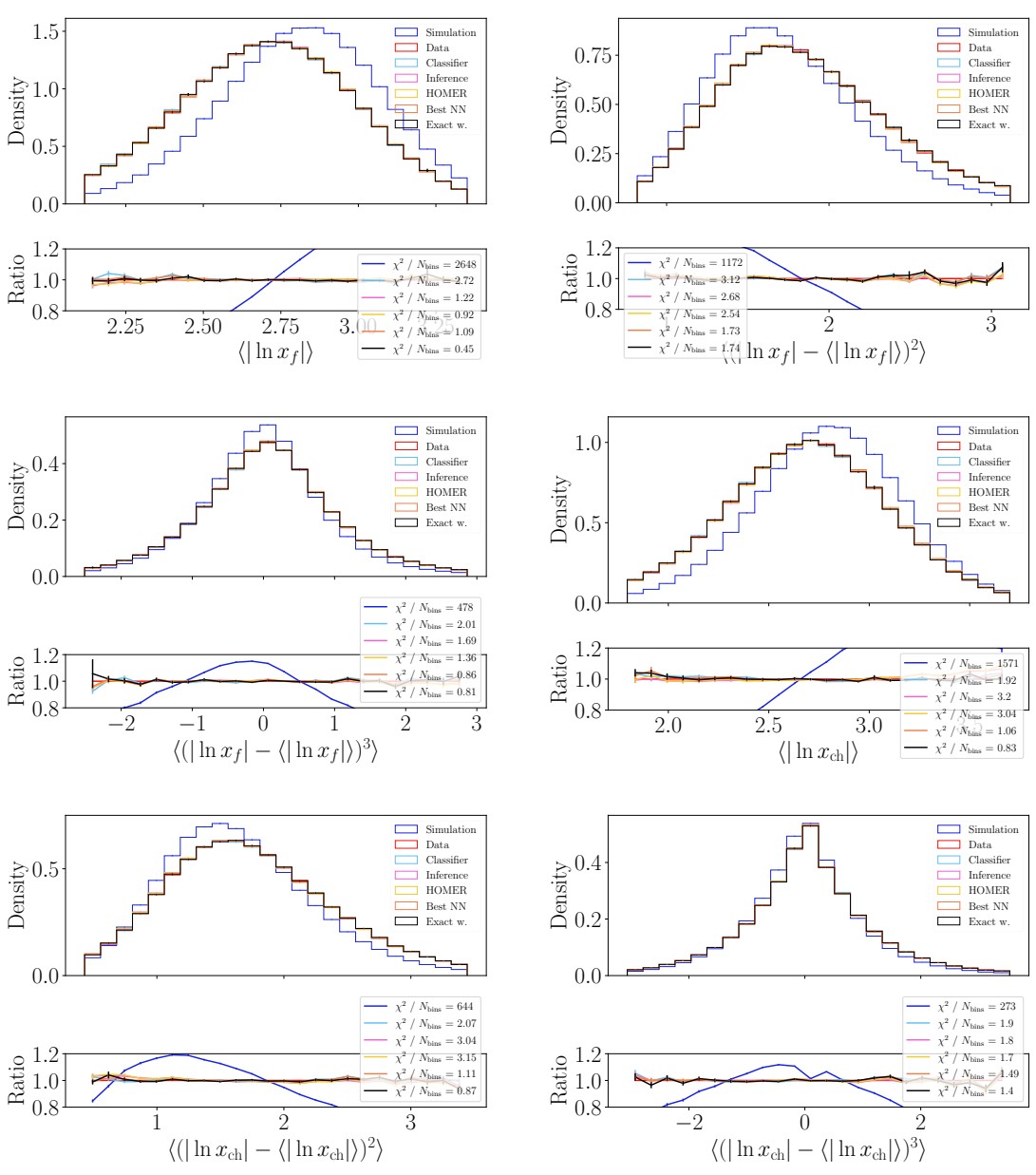

Figure 16: Distributions of the first three moments of $\ln x$, where $x = 2|\vec{p}|/\sqrt{s}$ for visible particle and charged particle distributions. Here, step 1 of the HOMER method is performed on unbinned high-level observables. See the main text for details.

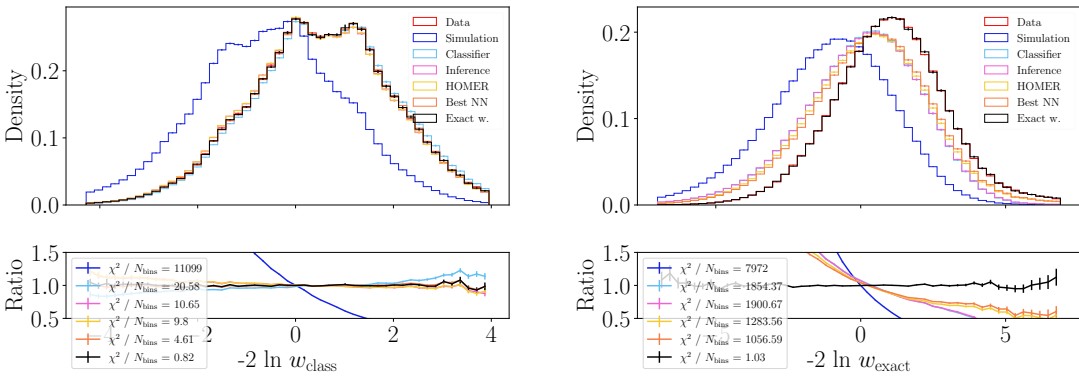

Figure 17: The distribution of the optimal observables (left) $-2\ln w_{\text{class}}$ and (right) $-2\ln w_{\text{exact}}$. The results are for a classifier trained on the unbinned high-level observables, see the text of section 3.1.2 for details.

in the left plot of fig. 21, where the $\chi^2/N_{\text{bins}}$ for the exact weights becomes smaller than for the unbinned high-level case shown in fig. 17. The cause of this decrease is a small fraction of events with large weights populating high-probability regions instead of the tails, increasing the uncertainty in those bins and thus reducing $\chi^2/N_{\text{bins}}$, while also sculpting the $w_{\text{class}}$ distribution.

Another diagnostic tool is the calibration curve of the classifier that compares the predicted positive ratio with the true positive ratio as a function of the classifier score. This curve, not shown here, yields worse results for the point cloud when compared to the case of training on unbinned high-level observables, with larger deviations from perfect calibration in the tails. The above observations indicate that there is room for improvement in the training of the point-cloud classifier, perhaps by increasing the number of training samples.

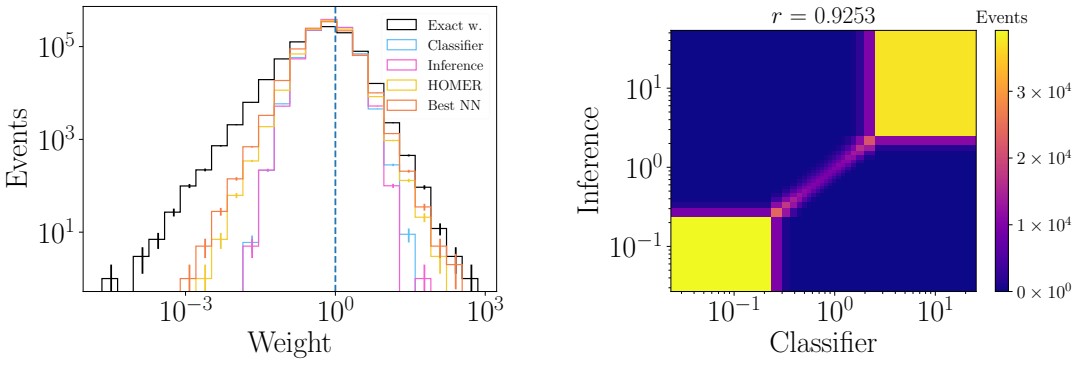

Figure 18: (left) The distributions of event weights $w(e_h)$ that follow from the HOMER method applied to the point cloud, section 3.2. (right) Comparison between $w_{\text{class}}(e_h)$ from step 1, on the $x$ axis, and $w_{\text{infer}}(e_h, \theta)$ from step 2, on the $y$ axis. The closer the Pearson correlation coefficient $r$ is to 1, the closer the match between $w_{\text{class}}(e_h)$ and $w_{\text{infer}}(e_h, \theta)$, signaling better training during step 2.



Figure 19: Distributions of high-level observables $1 - T$, $C$, $D$, $B_W$ and $B_T$, for definitions see appendix B, for the case where step 1 of the HOMER method is performed on the point cloud. See the main text for details.

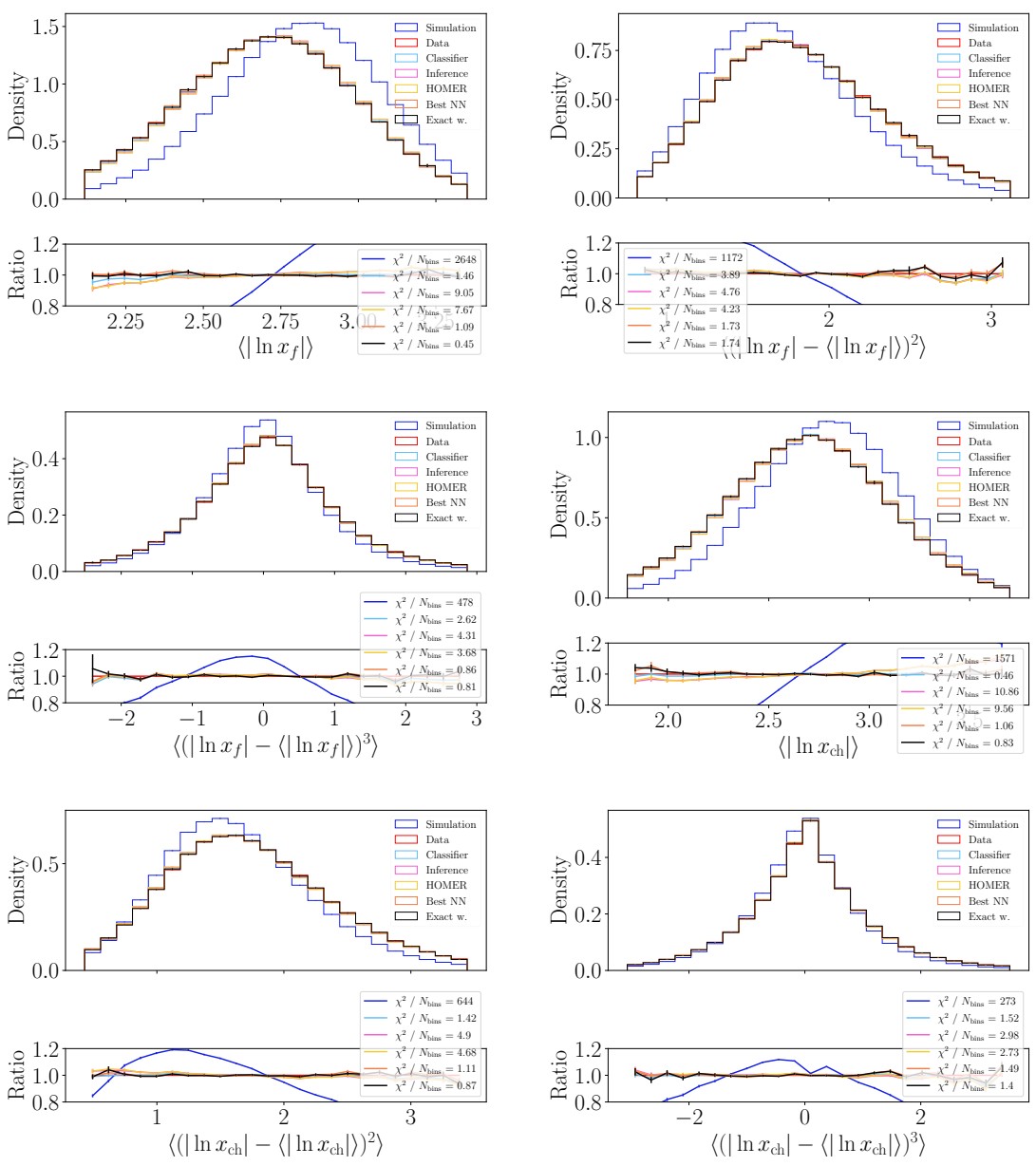

Figure 20: Distributions of the first three moments of $\ln x$, where $x = 2|\vec{p}|/\sqrt{s}$ for visible particle and charged particle distributions. Here, step 1 of the HOMER method is performed on point cloud. See the main text for details.

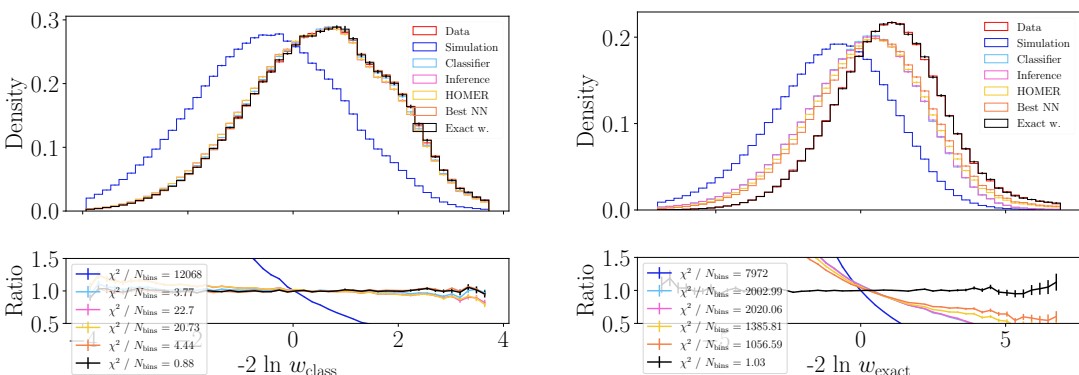

Figure 21: The distribution of the optimal observables (left) $-2\ln w_{\text{class}}$ and (right) $-2\ln w_{\text{exact}}$. The results are for a classifier trained on the point cloud, see the text of section 3.2 for details.

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
