# Peer review of "Describing Hadronization via Histories and Observables for Monte-Carlo Event Reweighting"

_SciPost Physics, doi:SciPost Phys. 18, 054 (2025)_

## Round 2 · Referee Report · Anonymous (Referee 1) · 2024-12-9

Strengths

1- Interesting development 2- ML of fragmentation function demonstrated 3- Conclusions beyond that demonstration 4- Written very clearly

Report

The authors report on a new method to extract the Lund symmetric fragmentation function in Pythia with the help of Machine Learning Algorithms. In the described HOMER (*H*istories and *O*bservables for *M*onte Carlo *E*vent *R*eweighting) method explizit weights are construced to classify events and to take into account the specifics of the fragmentation algorithm in Pythia. The method is applied to toy data for the fragmentation of a light quark string into pions only.

The results are very interesting and show that the fragmentation function can indeed be learned by a ML algorithm. While this already meets the goal of the paper there are some interesting conclusions drawn along the way. One particularly interesting exercise was that the ML methodology was applied once to a set conventional binned observables to describe e.g. event shapes. In addition event-by-event data were used in order to learn the fragmentation function with the result that not much improvement could be made, at least not for this simplified scenario.

The paper is written very clearly and the results are a very interesting step not only to demonstrate that ML methods are capable of reproducing the fragmentation function, the central ingredient of the Lund fragmentation model, but rather to show that even more detailed information will not give much added value. On the other hand a careful analysis of the likelihood distributions shows that the data does not ultimately constrain the fragmentation function but there would be more potential from a more detailed data set.

The fact that there is the possibility to draw conclusions like this shows that there is indeed some added value in exercises like this. While a detailed understanding of the model might even be hindered by ML methods, these may nonetheless point out new directions for model development.

The manuscript clearly meets the criteria SciPost and is recommended for publication.

Just one typo: p13, top line "distribution" -> "distributions"

Recommendation

Publish (easily meets expectations and criteria for this Journal; among top 50%)

  • validity: top
  • significance: top
  • originality: high
  • clarity: top
  • formatting: perfect
  • grammar: perfect

Author:  Manuel Szewc  on 2025-01-13  [id 5114]

(in reply to Report 1 on 2024-12-09)

We thank the reviewer for his/her positive report and have corrected the pointed out typographical error.

---

## Round 2 · Referee Report · Anonymous (Referee 2) · 2024-12-18

Strengths

  • the method can be used to train from data, improving PYTHIA
  • visualizations, like figs. 1 or 2 are well done.
  • formulas are explained well, term by term

Weaknesses

  • there might be a bias in the shown performance, as the plotted data was also used for model selection.

Report

Report on the manuscript "Describing Hadronization via Histories and Observables for Monte-Carlo Event Reweighting" by Christian Bierlich, Phil Ilten, Tony Menzo, Stephen Mrenna, Manuel Szewc, Michael K. Wilkinson, Ahmed Youssef, and Jure Zupan.

The manuscript describes a method (called HOMER) to extract a fragmentation model from experimental data, without requiring an explicit functional form. HOMER is a multi-step procedure that first learns to reweight simulation to data and then learns to reweight individual splittings to reproduce the learned data weight. In a final step, a fragmentation model can be extracted. The method uses intermediate information from simulation, but can be trained on experimental data to improve the simulation model beyond its current form.
I think the manuscript is very good, and it should definitively be published, as it easily meets the criteria. It is already very detailed and well written. I have only a few questions (mostly for my own curiosity) and minor suggestions that I would like to see addressed before.

Requested changes

  • In the description of step 3, I'm missing an explanation on how to extract $f_{data}$ from the learned $\omega$.
  • In section 3, the authors say they split the data in 2 parts, one for training, one for testing. The latter was used for both, verification of absence of overfitting and visualization of results. It would be better if the visualizations and evaluations would be based on a third, independent dataset, that was neither used for training nor model selection, as otherwise the presented results might be biased towards having selected the best model and not show the performance on unseen, independent data.
  • Figure 3 (and others in that style): the chosen binning is not very good. The first bin takes up 3/4 of the plot, and all the others are barely visible. Please consider using np.logspace or similar when defining the bins.
  • When discussing the final results (fig. 7), I was wondering: have you considered using symbolic regression on the result to see what functional form was extracted from data?
  • When using the point-cloud representation, you mentioned zero padding the event to size 100. This, in fact, means that you are not using a point cloud (of varying size), but instead just low-level observables as input. Have you considered using a Deep Sets model or anything else that allows for differently sized input data? Please consider rephrasing the sections to reflect that you use fixed-size inputs.

Minor comments: - in the paragraph just above section 2.2.2, there is a typo: "expect" -> "except" - In section 3.1 "high level observables" and "high-level observables" are both used. Please use "high-level observables" consistently. - In appendix C.1, in the sentence "There is a significant reduction in $\chi^2/N_{bins}$ ..., unless these were already captured well by the unbinned training.", shouldn't the last part be "binned training"?

Recommendation

Ask for minor revision

  • validity: top
  • significance: top
  • originality: top
  • clarity: top
  • formatting: perfect
  • grammar: perfect

Author:  Manuel Szewc  on 2025-01-13  [id 5113]

(in reply to Report 2 on 2024-12-18)

We thank the reviewer for constructive comments. Below we address in detail the issues raised by the reviewer, and list the corresponding changes made to the manuscript.

Requested changes.

  • We have added a more detailed explanation in section 2.2.3 on how one obtains $f_{\mathrm{data}}$, which now also complements the discussion in section 3.1.1, where we describe how fig. 7 was obtained.
  • The reviewer is correct in that there is a danger of biasing our results when using the same dataset to assess overfitting and display the final results. We have added a warning to this effect in section 3, recommending the use of three datasets in case of more realistic applications. However, we do believe that the addition of the third dataset is not necessary for the purposes of this manuscript, i.e., for the demonstration of the proof of concept. Since we are performing a closure test, the effect of possible over-fitting to the test dataset can be directly assessed from our numerical analysis, given that we have full knowledge of the correct data distributions, and was found to be minimal.
  • We thank the reviewer for the suggestion. We have changed the binning to logarithmic and find that it indeed increases the clarity of the plots.
  • The reviewer makes a very insightful question. We have considered such a study but have left it for future work. We did perform a more limited test where we perform a parametric fit to the learned fragmentation function and have verified that the obtained data-driven fragmentation function is consistent with the Lund fragmentation function with the right parameters. It would be very interesting to confirm our closure test through symbolic regression and furthermore, to eventually explore what one can learn from the actual experimental data.
  • The reviewer is correct in pointing out that a point cloud consists of collections of varying size events. We agree with that definition and our events do have varying multiplicity and thus fit that definition, which is why we name it the point-cloud representation. The mentioned zero-padding and corresponding fixed size is just implemented to store the generated events. The Step 1 classifier for the point cloud classifier is a Deep Set model that takes differently sized input data. To train and infer using this model, we go from the stored fixed size events to the point-cloud representation using masking. We have clarified this in 3.2.

Minor comments: * Fixed. * Fixed. * Fixed.

---

## Round 3 · Referee Report · Anonymous (Referee 2) · 2025-1-16

Report

The authors have addressed all my previous comments. I recommend this manuscript for publication in SciPost Physics.

Recommendation

Publish (surpasses expectations and criteria for this Journal; among top 10%)

---

## Round 3 · Author Response

We thank the reviewers for constructive comments. Below we address in detail the issues raised, and list the corresponding changes made to the manuscript.

---

## Round 3 · List of Changes

• We have added a more detailed explanation in section 2.2.3 on how one obtains the measured fragmentation function, which now also complements the discussion in section 3.1.1, where we describe how fig. 7 was obtained.
  • We have added a warning regarding a source of possible overfitting in section 3, recommending the use of three datasets in case of more realistic applications.
  • We have changed the binning of fig. 3 and others in that style to logarithmic.
  • We have clarified our definition of point cloud and the use of a Deep Sets-based classifier in section 3.2.
  • We have fixed the typos pointed out by the reports.

---

## Editorial Decision

published